# Avoiding Robust Misclassifications for Improved Robustness without Accuracy Loss

## Abstract

While current methods for training robust deep learning models optimize robust accuracy, in practice, the resulting models are often both robust and inaccurate on numerous samples, providing a false sense of safety for those. Further, they significantly reduce natural accuracy, which hinders the adoption in practice. In this work, we address both of these challenges by extending prior works in three main directions. First, we propose a new training method that jointly maximizes robust accuracy and minimizes robust inaccuracy. Second, since the resulting models are trained to be robust only if they are accurate, we leverage robustness as a principled abstain mechanism. Finally, this abstain mechanism allows us to combine models in a compositional architecture that significantly boosts overall robustness without sacrificing accuracy. We demonstrate the effectiveness of our approach to both empirical and certified robustness on six recent state-of-the-art models and using several datasets. For example, on CIFAR-10 with $\varepsilon_\infty = 1/255$, it successfully enhanced the robust accuracy of a state-of-the-art standard trained model from 26% to 86% while only marginally reducing its natural accuracy from 97.8% to 97.6%.

## 1 Introduction

In recent years, there has been a significant amount of work that studies and improves both adversarial (Szegedy et al., 2013; Goodfellow et al., 2014; Carlini & Wagner, 2017; Croce & Hein, 2020; Madry et al., 2018) and certified robustness (Balunovic & Vechev, 2019; Cohen et al., 2019; Salman et al., 2019; Xu et al., 2020; Zhai et al., 2020; Zhang et al., 2019b) of neural networks. However, currently, there are two key limitations that hinder the wider adoption of robust models in practice.

**Existing Models are Robustly Inaccurate** First, despite substantial progress in training robust models, existing works usually only report robust accuracy, i.e., samples for which the model robustly predicts the correct label. Meanwhile, the issue of robust inaccuracy, i.e., samples that are robustly misclassified with a wrong label, is usually not even reported (we formally define robust inaccuracy in Equation 3). This is especially problematic for safety-critical models, where the robustness can be mistakenly used as a safety argument. We quantify the severity of this issue in Table 1, by evaluating recent state-of-the-art robust models. As can be seen, recent models contain up to 15% of robust inaccurate samples and the ratio of such samples worsens with smaller perturbation regions.

| | CIFAR-10 Krizhevsky et al. | | | CIFAR-100 Krizhevsky et al. | MTSD Appendix A.1 | SBB Appendix A.1 |
|---|---|---|---|---|---|---|
| | Zhang et al. | Carmon et al. | Gowal et al. | Rebuffi et al. | Zhang et al. | Zhang et al. |
| $\mathcal{B}^\infty_{1/255}$ | 4.6% | 3.6% | 2.9% | 15.2% | 3.9% | 7.0% |
| $\mathcal{B}^\infty_{4/255}$ | 3.3% | 1.1% | 0.9% | 4.3% | 2.3% | 7.1% |
| $\mathcal{B}^\infty_{8/255}$ | 2.6% | 0.8% | 1.3% | 3.9% | 1.7% | 6.4% |

Table 1: Percentage of robust and inaccurate samples for various recent robust models and datasets, which we describe in more detail in Section 7. Each model is trained for the indicated threat model and evaluated using 40-step APGD (Croce & Hein, 2020).

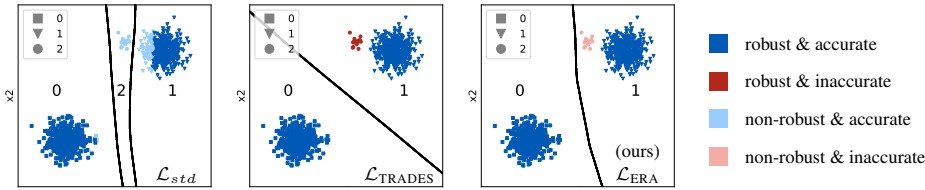

Figure 1: Decision regions for models trained via standard training $\mathcal{L}_{std}$, adversarial training $\mathcal{L}_{\text{TRADES}}$ (Zhang et al., 2019a), and our training $\mathcal{L}_{\text{ERA}}$ (Equation 5). In this case, our training achieves the same robust accuracy as $\mathcal{L}_{\text{TRADES}}$ but avoids all robust inaccurate samples by making them non-robust. Note that all three models predict over all three classes, however, the decision regions for class 2 of the $\mathcal{L}_{\text{TRADES}}$ and $\mathcal{L}_{\text{ERA}}$ trained models happen to be too small to be visible. The considered model architecture and the training hyperparameters are provided in Appendix A.2.

**Robustness vs Accuracy Tradeoff**   Second, existing robust training methods improve the model robustness, but they also typically degrade the standard accuracy on unperturbed inputs. To address this limitation, a number of recent works study this issue in detail and propose new methods to mitigate it (Mueller et al., 2020; Raghunathan et al., 2020; Yang et al., 2020; Stutz et al., 2019).

**Our Work**   In this work, we advance the line of work that aims to boost robustness without sacrificing accuracy, but we approach the problem from a new perspective – by avoiding robust inaccuracy.

Concretely, we propose a new training method that jointly maximizes robust accuracy while minimizing robust inaccuracy. We illustrate the effect of our training on a synthetic dataset (described in Appendix A.2) in Figure 1, showing the decision boundaries of three models, trained using standard training $\mathcal{L}_{std}$, adversarial training $\mathcal{L}_{\text{TRADES}}$ (Zhang et al., 2019a), and our training $\mathcal{L}_{\text{ERA}}$ (Equation 5). First, observe that while the $\mathcal{L}_{std}$ trained model achieves $100\%$ accuracy, only $91.1\%$ of these samples are robust (and accurate). When using $\mathcal{L}_{\text{TRADES}}$, we can observe the robustness vs accuracy tradeoff – the robust accuracy improves to $98.4\%$ at the expense of $1.6\%$ (robust) inaccuracy. In contrast, using our $\mathcal{L}_{\text{ERA}}$, we retain the high robust accuracy of $98.4\%$ but avoid all robust inaccurate samples by appropriately shifting the decision boundary, rendering them non-robust.

Second, since our models are trained to be robust only if they are accurate, we leverage robustness as a principled abstain mechanism. This abstain mechanism then allows us to combine models in a compositional architecture that significantly boosts overall robustness without sacrificing accuracy. Concretely, in Figure 1, we would define a selector model that abstains on all non-robust samples. Then, the abstained (non-robust) samples are evaluated by the standard trained model $\mathcal{L}_{std}$, while the selected samples are evaluated using the robust model $\mathcal{L}_{\text{ERA}}$. This allows us to achieve the best of both models – high robust accuracy ($98.4\%$), high natural accuracy ($100\%$), and no robust inaccuracy.

We show the practical effectiveness of our approach by instantiating over several datasets and existing robust models for both empirical and certified robustness. Our evaluations show that our method effectively reduces robust and inaccurate samples by up to 97.28%. Further, our approach significantly improves robustness for CIFAR-10, CIFAR-100, MTSD and SBB datasets by 60.3%, 38.8%, 29.2% and 37.7%, respectively, while simultaneously decreasing natural accuracy by only 0.2%.

## 2   RELATED WORK

There is a growing body of work that extends models with an abstain option, either by training a selector mechanism separately or jointly together with the model. The existing approaches include various selection mechanisms such as entropy selection (Mueller et al., 2020), selection function (Cortes et al., 2016; Mueller et al., 2020; Geifman & El-Yaniv, 2019), softmax response (Stutz et al., 2020; Geifman & El-Yaniv, 2017), or explicit abstain class (Laidlaw & Feizi, 2019; Liu et al., 2019). In our work, we explore an alternative selector mechanism that uses model robustness. The advantage of this formulation is that the selector provides strong guarantees for each sample and can never produce false-positive selections. The disadvantage is that it introduces a significant runtime overhead, compared to many other methods that require only a single forward pass.

At the same time, several recent works investigated the robustness and accuracy tradeoff both theoretically (Yang et al., 2020; Dobriban et al., 2020) and practically by proposing new methods to mitigate it. For example, Raghunathan et al. (2020) proposes robust self-training that leverages unlabeled data to regularize the model. Stutz et al. (2019) considers a new method based on on-manifold adversarial examples, which are more aligned with the true data distribution than the $\ell_p$-norm noise models. Mueller et al. (2020) focuses on deterministic certification and also proposes using compositional models to control the robustness and accuracy tradeoff. In our work, we also take advantage of compositional models, but we focus on both empirical and probabilistic certified robustness. Further, our selector formulation is based on a new training that minimizes robust inaccuracy and can be used to fine-tune any existing robust model. Finally, we provide individual robustness at inference time, rather than distributional robustness considered in prior works.

## 3 PRELIMINARIES

Let $f_\theta : \mathbb{R}^d \to \mathbb{R}^k$ be a neural network which classifies inputs $\boldsymbol{x} \in \mathcal{X} \subseteq \mathbb{R}^d$ to outputs $\mathbb{R}^k$ (e.g., logits or probabilities). The hard classifier induced by the network is given as $F_\theta(\boldsymbol{x}) = \arg\max_{i \in \mathcal{Y}} f_\theta(\boldsymbol{x})_i$, where $f_\theta(\boldsymbol{x})_i$ is the network output for the $i$-th class and $\mathcal{Y}, |\mathcal{Y}| = k$ is the finite set of discrete labels.

**Natural Accuracy** Given a distribution over input-label pairs $\mathcal{D}$ and a classifier $F_\theta : \mathcal{X} \to \mathcal{Y}$, an input-label pair $(\boldsymbol{x}, y)$ is considered accurate iff the classifier $F_\theta$ predicts the correct label $y$ for $\boldsymbol{x}$:

$$\mathcal{R}_{nat}(F_\theta) = \mathbb{E}_{(\boldsymbol{x},y) \sim \mathcal{D}} \quad \mathbf{1}\{F_\theta(\boldsymbol{x}) = y\} \tag{1}$$

**Robust Accuracy** Given an input-label pair $(\boldsymbol{x}, y)$, we say that the classifier $F_\theta$ is robust and accurate iff it predicts the correct label $y$ for all samples from a predefined region $\mathcal{B}^p_\varepsilon(\boldsymbol{x})$, such as a $\ell_p$-norm ball centered at $\boldsymbol{x}$ with radius $\varepsilon$, i.e., $\mathcal{B}^p_\varepsilon(\boldsymbol{x}) := \{\boldsymbol{x}' : ||\boldsymbol{x}' - \boldsymbol{x}||_p \leq \varepsilon\}$. Formally:

$$\mathcal{R}^{acc}_{rob}(F_\theta) = \mathbb{E}_{(\boldsymbol{x},y) \sim \mathcal{D}} \quad \mathbf{1}\{F_\theta(\boldsymbol{x}) = y\} \wedge \mathbf{1}\{\forall \boldsymbol{x}' \in \mathcal{B}^p_\varepsilon(\boldsymbol{x}). \ F_\theta(\boldsymbol{x}') = F_\theta(\boldsymbol{x})\} \tag{2}$$

**Robust Inaccuracy** Similarly to robust accuracy, an input-label pair $(\boldsymbol{x}, y)$ is considered robustly inaccurate iff the classifier $F_\theta$ predicts an incorrect label $F_\theta(\boldsymbol{x}) \neq y$ and $F_\theta$ is robust towards that misprediction for all inputs in $\mathcal{B}^p_\varepsilon(\boldsymbol{x})$. Formally, the robust inaccuracy is defined as:

$$\mathcal{R}^{\neg acc}_{rob}(F_\theta) = \mathbb{E}_{(\boldsymbol{x},y) \sim \mathcal{D}} \quad \mathbf{1}\{F_\theta(\boldsymbol{x}) \neq y\} \wedge \mathbf{1}\{\forall \boldsymbol{x}' \in \mathcal{B}^p_\varepsilon(\boldsymbol{x}). \ F_\theta(\boldsymbol{x}') = F_\theta(\boldsymbol{x})\} \tag{3}$$

## 4 REDUCING ROBUST INACCURACY: ADVERSARIAL & CERTIFIED TRAINING

In this section, we present our training method that extends existing robust training approaches by also considering samples that are robust but inaccurate. We start by describing a high-level problem statement which we then instantiate for both empirical robustness as well as certified robustness.

**Problem Statement** Given a distribution over input-label pairs $\mathcal{D}$, our goal is to find model parameters $\theta$ such that the resulting model maximizes robust accuracy, while at the same time minimizing robust inaccuracy. Concretely, this translates to the following optimization objective:

$$\arg\min_\theta \mathbb{E}_{(\boldsymbol{x},y) \sim \mathcal{D}} \quad \underbrace{\beta \cdot \mathcal{L}_{rob}(\boldsymbol{x}, y)}_{\text{optimize robust accuracy}} + \underbrace{\mathbf{1}\{F_\theta(\boldsymbol{x}) \neq y\} \cdot \mathcal{L}^{\neg acc}_{rob}(\boldsymbol{x}, y)}_{\text{penalize robust inaccuracy}} \tag{4}$$

where $\beta \in \mathbb{R}^+$ is a regularization term, $\mathbf{1}\{F_\theta(\boldsymbol{x}) \neq y\}$ is an indicator function denoting samples for which the model is inaccurate, and $\mathcal{L}_{rob}(\boldsymbol{x}, y)$ with $\mathcal{L}^{\neg acc}_{rob}(\boldsymbol{x}, y)$ are loss functions that optimize robust accuracy and penalize robust inaccuracy, respectively. Here, the first loss function $\mathcal{L}_{rob}(\boldsymbol{x}, y)$ is standard and can be directly instantiated using existing approaches (see next). The main challenge comes in defining the second loss term, as well as ensuring that the resulting formulation is easy to optimize, e.g., by defining a smooth approximation of the non-differentiable indicator function.

### 4.1 ADVERSARIAL TRAINING

We instantiate the loss function from Equation 4 when training empirically robust models as follows:

$$\mathcal{L}_{\text{ERA}}(f_\theta, (\boldsymbol{x}, y)) = \beta \cdot \mathcal{L}_{\text{TRADES}}(f_\theta, (\boldsymbol{x}, y)) + (1 - f_\theta(\boldsymbol{x})_y) \min_{\boldsymbol{x}' \in \mathcal{B}^p_\varepsilon(\boldsymbol{x})} \ell_{\text{CE}}(f_\theta(\boldsymbol{x}'), \arg\max_{c \in \mathcal{Y} \setminus \{F_\theta(\boldsymbol{x})\}} f_\theta(\boldsymbol{x}')_c)$$

$$\tag{5}$$

Below, we introduce each term in more detail and discuss the motivation behind our formulation.

$\mathcal{L}_{rob}$   To instantiate $\mathcal{L}_{rob}$, we can use any existing adversarial training method. For example, considering standard adversarial training (Goodfellow et al., 2014) would result in the following loss:

$$\mathcal{L}_{adv} := \max_{\boldsymbol{x}' \in \mathcal{B}_{\varepsilon}^p(\boldsymbol{x})} \ell_{\text{CE}}(f_\theta(\boldsymbol{x}'), y) \tag{6}$$

where $\ell_{\text{CE}}$ is cross-entropy loss of the worst example in the allowed perturbation region $\mathcal{B}_{\varepsilon}^p$. Similarly, we can instantiate the loss using more sophisticated methods, such as TRADES (Zhang et al., 2019a):

$$\mathcal{L}_{\text{TRADES}} := \ell_{\text{CE}}(f_\theta(\boldsymbol{x}), y) + \gamma \max_{\boldsymbol{x}' \in \mathcal{B}_{\varepsilon}^p(\boldsymbol{x})} D_{\text{KL}}(f_\theta(\boldsymbol{x}), f_\theta(\boldsymbol{x}')) \tag{7}$$

where $D_{\text{KL}}$ is the Kullback-Leibler divergence (Kullback & Leibler, 1951).

$\mathbf{1}\{F_\theta(\boldsymbol{x}) \neq y\}$   Next, we consider the indicator function, which encourages learning on inaccurate samples. Since the indicator function is computationally intractable, we replace the hard qualifier by a soft qualifier $1 - f_\theta(\boldsymbol{x})_y$. The soft qualifier will be small for accurate and large for inaccurate samples, thus providing a smooth approximation of the original indicator function.

$\mathcal{L}_{rob}^{\neg acc}$   Third, we define the loss that penalizes robust but inaccurate samples. This can be formulated similar to the adversarial training objective (Madry et al., 2018), however, instead of optimizing the prediction of the adversarial example $f_\theta(\boldsymbol{x}')$ towards the correct label $y$, we optimize towards the most likely adversarial label $\arg\max_{c \in \mathcal{Y} \setminus \{F_\theta(\boldsymbol{x})\}} f_\theta(\boldsymbol{x}')_c$. This leads to the following formulation:

$$\min_{\boldsymbol{x}' \in \mathcal{B}_{\varepsilon}^p(\boldsymbol{x})} \ell_{\text{CE}}(f_\theta(\boldsymbol{x}'), \arg\max_{c \in \mathcal{Y} \setminus \{F_\theta(\boldsymbol{x})\}} f_\theta(\boldsymbol{x}')_c) \tag{8}$$

Note that the purpose of the $\mathcal{L}_{rob}^{\neg acc}$ loss is to penalize robustness by making the model non-robust. As a result, it is sufficient to consider only a single non-robust example, thus the minimization (rather than maximization) in the loss objective[1].

## 4.2   CERTIFIED TRAINING

Similarly to Section 4.1, we now instantiate the loss function from Equation 4 for probabilistic certified robustness via randomized smoothing (Cohen et al., 2019). Randomized smoothing constructs a smoothed classifier $G_\theta \colon \mathcal{X} \to \mathcal{Y}$ from a base classifier $F_\theta$, where $G_\theta(\boldsymbol{x})$ predicts the class which $F_\theta$ is most likely to return when $\boldsymbol{x}$ is perturbed under isotropic Gaussian noise. Our proposed instantiation of Equation 4 for probabilistic certified robustness is as follows:

$$\mathcal{L}_{\text{CRA}}(f_\theta, (\boldsymbol{x}, y)) = \beta \cdot \mathcal{L}_{noise}(f_\theta, (\boldsymbol{x}, y)) + \frac{1}{k} \sum_{j=1}^{k} \left(1 - f_\theta(\boldsymbol{x} + \boldsymbol{\eta}_j)_y\right) CR(f_\theta, (\boldsymbol{x}, y)) \tag{9}$$

where $\boldsymbol{\eta}_1, ..., \boldsymbol{\eta}_k$ are $k$ i.i.d. samples from $\mathcal{N}(0, \sigma^2 \boldsymbol{I})$. Note that, since the robustness guarantees provided by randomized smoothing hold for the smoothed classifier $G_\theta$, the three loss components from Equation 4 need to be formulated with respect to the smoothed classifier $G_\theta$.

$\mathcal{L}_{rob}$   To instantiate $\mathcal{L}_{rob}$, we can use any existing certified training method for randomized smoothing, such as the methods defined by Cohen et al. (2019) or Zhai et al. (2020). Concretely, when using Cohen et al. (2019), the loss is defined using Gaussian noise augmentation:

$$\mathcal{L}_{noise} := \ell_{\text{CE}}(f_\theta(\boldsymbol{x} + \boldsymbol{\eta}), y), \qquad \boldsymbol{\eta} \sim \mathcal{N}(0, \sigma^2 \boldsymbol{I}) \tag{10}$$

$\mathbf{1}\{F_\theta(\boldsymbol{x}) \neq y\}$   We again replace the computationally intractable hard qualifier by a soft qualifier $\mathbb{E}_{\boldsymbol{\delta} \sim \mathcal{N}(0, \sigma^2 \boldsymbol{I})}[1 - f_\theta(\boldsymbol{x} + \boldsymbol{\delta})_y]$, which encodes the misprediction probability of the smoothed classifier. In practice, we approximate expectations over Gaussians via Monte Carlo sampling, thus leading to the approximated soft inaccuracy qualifier $1/k \sum_{j=1}^{k} 1 - f_\theta(\boldsymbol{x} + \boldsymbol{\eta}_j)_y$.

---

[1]Naturally, this assumes that the method used to check robustness can correctly detect the non-robustness, even if it is caused by a single example. Note that, for a fair evaluation, we use a relatively weak 10-step PGD (Madry et al., 2018) attack during training and a strong 40-step APGD (Croce & Hein, 2020) for evaluation.

$\mathcal{L}_{rob}^{\neg acc}$   Finally, we instantiate the $\mathcal{L}_{rob}^{\neg acc}$ loss term, which encourages the model toward non-robust predictions on robust but inaccurate samples. We propose to minimize robustness by directly minimizing the certified radius of the smoothed classifier $G_\theta$. The certified radius formulation by Cohen et al. (2019) involves a sum of indicator functions, which is not differentiable. However, Zhai et al. (2020) have recently proposed the following differentiable certified radius formulation:

$$CR(f_\theta, (\boldsymbol{x}, y)) = \frac{\sigma}{2} \Big[ \Phi^{-1} \big( \frac{1}{k} \sum_{j=1}^{k} f_\theta(\boldsymbol{x} + \boldsymbol{\eta}_j; \Gamma)_y \big) - \Phi^{-1} \big( \max_{y' \neq y} \frac{1}{k} \sum_{j=1}^{k} f_\theta(\boldsymbol{x} + \boldsymbol{\eta}_j; \Gamma)_{y'} \big) \Big] \quad (11)$$

where $\Phi^{-1}$ is the inverse of the standard Gaussian CDF, $\Gamma$ is the inverse softmax temperature multiplied with the logits of $f_\theta$, and $\boldsymbol{\eta}_{1:k}$ are $k$ *i.i.d.* samples from $\mathcal{N}(0, \sigma^2 \boldsymbol{I})$. Note that, by setting the loss term $\mathcal{L}_{rob}^{\neg acc}$ to $CR(f_\theta, (\boldsymbol{x}, y))$, we directly penalize robustness of the smoothed classifier $G_\theta$.

## 5   ROBUST ABSTAIN MODELS

In this section, we extend the models trained so far by leveraging robustness as a principled abstain mechanism. Further, we define an additional loss function based on the Deep Gamblers loss (Liu et al., 2019), which is specifically designed for training adversarially robust abstain models.

**Abstain Model**   Consider an input space $\mathcal{X} \subseteq \mathbb{R}^d$ and a label space $\mathcal{Y}$. A model with an abstain option (El-Yaniv et al., 2010) is a pair of functions $(F_\theta, S)$, where $F_\theta \colon \mathcal{X} \to \mathcal{Y}$ is a classifier and $S \colon \mathcal{X} \to \{0, 1\}$ is a selection mechanism, which acts as a binary qualifier for $F_\theta$. Let $S(\boldsymbol{x}) = 0$ indicate that the model abstains on input $\boldsymbol{x} \in \mathcal{X}$, while $S(\boldsymbol{x}) = 1$ indicates that the model commits to the classifier $F_\theta$ for input $\boldsymbol{x}$ and predicts $F_\theta(\boldsymbol{x})$.

**Robustness Indicator Selector**   We now instantiate abstain models with a robustness indicator selector, that abstains on all non-robust samples. For adversarial robustness, the selector is defined as:

$$S_{\texttt{ERI}}(\boldsymbol{x}) = \mathbf{1}\{\forall \boldsymbol{x}' \in \mathcal{B}(\boldsymbol{x}) \colon F_\theta(\boldsymbol{x}') = F_\theta(\boldsymbol{x})\} \quad (12)$$

For certified robustness, the selector is defined as:

$$S_{\texttt{CRI}}(\boldsymbol{x}) = \mathbf{1}\{\forall \boldsymbol{x}' \in \mathcal{B}(\boldsymbol{x}) \colon G_\theta(\boldsymbol{x}') = G_\theta(\boldsymbol{x})\} \quad (13)$$

**Robustness Guarantees: Robust Selection**   Similar to robust accuracy, the robustness of an abstain model needs to be evaluated with respect to a threat model. In our work, we consider the same threat model as for the underlying model $F_\theta$, namely $\mathcal{B}_\varepsilon^p(\boldsymbol{x}) := \{\boldsymbol{x}' \colon ||\boldsymbol{x}' - \boldsymbol{x}||_p \leq \varepsilon\}$, a $\ell_p$-norm ball centered at $\boldsymbol{x}$ with radius $\varepsilon$. Then, we define the robust selection of an abstain model as follows:

$$\mathcal{R}_{rob}^{sel}(S) = \mathbb{E}_{(\boldsymbol{x}, y) \sim \mathcal{D}} \quad \mathbf{1}\{\forall \boldsymbol{x}' \in \mathcal{B}_\varepsilon^p(\boldsymbol{x}). \, S(\boldsymbol{x}') = 1\} \quad (14)$$

That is, we say that a model is robustly selecting $\boldsymbol{x}$ if the selector $S$ would select all valid perturbations $\boldsymbol{x}' \in \mathcal{B}_\varepsilon^p(\boldsymbol{x})$. When used together with our definition of $S_{\texttt{ERI}}$, we obtain the following criterion:

$$\begin{aligned} \mathcal{R}_{rob}^{sel}(S_{\texttt{ERI}}) &= \mathbb{E}_{(\boldsymbol{x}, y) \sim \mathcal{D}} \quad \mathbf{1}\{\forall \boldsymbol{x}' \in \mathcal{B}_\varepsilon^p(\boldsymbol{x}). \, S_{\texttt{ERI}}(\boldsymbol{x}') = 1\} \\ &= \mathbb{E}_{(\boldsymbol{x}, y) \sim \mathcal{D}} \quad \mathbf{1}\{\forall \boldsymbol{x}' \in \mathcal{B}_\varepsilon^p(\boldsymbol{x}). \, \mathbf{1}\{\forall \boldsymbol{x}'' \in \mathcal{B}_\varepsilon^p(\boldsymbol{x}'). \, F_\theta(\boldsymbol{x}'') = F_\theta(\boldsymbol{x}')\}\} \\ &= \mathbb{E}_{(\boldsymbol{x}, y) \sim \mathcal{D}} \quad \mathbf{1}\{\forall \boldsymbol{x}' \in \mathcal{B}_{2 \cdot \varepsilon}^p(\boldsymbol{x}). \, F_\theta(\boldsymbol{x}') = F_\theta(\boldsymbol{x})\} \end{aligned}$$

In other words, to guarantee that the selector $S_{\texttt{ERI}}$ is robust for all $\boldsymbol{x}' \in \mathcal{B}_\varepsilon^p(\boldsymbol{x})$, we in fact need to check robustness of the model $F_\theta$ to double that region $\boldsymbol{x}' \in \mathcal{B}_{2 \cdot \varepsilon}^p(\boldsymbol{x})$. This is important in order to obtain the correct guarantees and is reflected in our evaluation in Section 7.

Note that when evaluating robust selection for certified training, it is sufficient to show that the smoothed model $G_\theta$ can be certified with a radius $R \geq \varepsilon$. Then, the smoothed model guarantees that $G_\theta(x') = c_A$ for all $\boldsymbol{x}' \in \mathcal{B}_\varepsilon^p(\boldsymbol{x})$, which is equivalent to our condition $\forall \boldsymbol{x}' \in \mathcal{B}(\boldsymbol{x}) \colon G_\theta(\boldsymbol{x}') = G_\theta(\boldsymbol{x})$.

## 6   BOOSTING ROBUSTNESS WITHOUT ACCURACY LOSS

Consider an abstain model $(F_\theta, S)$ and let $\mathcal{D}$ be a dataset to evaluate $(F_\theta, S)$. The selector $S$ partitions $\mathcal{D}$ into two disjoint subsets – the set of abstained inputs $\mathcal{D}_{\neg s}$ and the set of selected inputs $\mathcal{D}_s$ for

which $F_\theta$ makes a prediction. However, depending on the application, it may be desirable to make a best-effort prediction on all samples, including $\mathcal{D}_{\neg s}$. This insight leads to compositional architectures, already used by a number of prior works (Mueller et al., 2020; Wong et al., 2018).

Let $H = ((F_{robust}, S), F_{core})$ be a 2-compositional architecture consisting of a selection mechanism $S$, a robustly trained model $F_{robust}$, and a core model $F_{core}$. Given an input $\boldsymbol{x} \in \mathcal{X}$, the selector $S$ decides whether the model is confident on $\boldsymbol{x}$ and commits to the robust model $F_{robust}$ or whether the model should abstain and fall back to the core model $F_{core}$. Formally:

$$H(\boldsymbol{x}) = S(\boldsymbol{x}) \cdot F_{robust}(\boldsymbol{x}) + (1 - S(\boldsymbol{x})) \cdot F_{core}(\boldsymbol{x}) \tag{15}$$

Note that each model of a compositional architecture can be chosen arbitrarily, regardless of the model's network architecture or its training. However, one benefit of using a compositional architecture is combining models that complement each other, resulting in overall improved model performance. In our work, we combine models trained via adversarial or certified training (which typically have lower natural accuracy), with models trained using standard training (which have high natural accuracy but low robustness). The compositional model performance then depends on the quality of the selector $S$, used to determine which model to evaluate for each sample.

## 7 EVALUATION

In this section, we present a thorough evaluation of our approach by instantiating it to four different datasets, six recent state-of-the-art models, for both adversarial and certified robustness, and including top-trained models from RobustBench (Croce et al., 2020). We show the following key results:

- Fine-tuning existing models with our proposed loss successfully decreases robust inaccuracy and provides a Pareto front of models with different robustness tradeoffs.
- Combining our proposed loss and robustness as an abstain mechanism leads to higher robust selection and accuracy compared to softmax response and selection network baselines.
- Our 2-compositional models significantly improve robustness by up to $+60\%$ while retaining the natural accuracy, causing only minor decrease of up to $-0.2\%$ (for $\mathcal{B}^\infty_{1/255}$ and $\mathcal{B}^\infty_{2/255}$).

We perform all experiments on a single GeForce RTX 3090 GPU and use PyTorch (Paszke et al., 2019) for our implementation. The hyperparameters used for our experiments are provided in Appendix A.2.

**Models** Our proposed training method requires neither retraining classifiers from scratch nor modifications to existing classifiers, thus our approach can be applied to fine-tune a wide range of existing models[2]. To demonstrate this, we use the following robust pre-trained models:

For *empirical robustness*, we evaluate five existing models from Carmon et al. (2019), Gowal et al. (2020), Rebuffi et al. (2021), Sehwag et al. (2021) and Zhang et al. (2019a), where all but the last model are top models taken from RobustBench (Croce et al., 2020). The model by Sehwag et al. is trained for $\varepsilon_2 = 0.5$, while the other models are trained for $\varepsilon_\infty = {}^8/{}_{255}$. In our evaluation, we fine-tune each model for 50 epochs for the considered threat model ($\varepsilon_\infty \in \{{}^1/{}_{255}, {}^2/{}_{255}, {}^4/{}_{255}\}$ and $\varepsilon_2 \in \{0.12, 0.25\}$) using both the $\mathcal{L}_{\text{TRADES}}$ loss (Zhang et al., 2019a) and our proposed $\mathcal{L}_{\text{ERA}}$ loss.

For *certified robustness*, we use the $\sigma = 0.12$ Gaussian noise augmentation trained model by Cohen et al. (2019) and the $\varepsilon_2 = 0.50$ adversarially trained model by Sehwag et al. (2021). Similarly to empirical robustness, we fine-tune the models for 50 epochs using either Gaussian noise augmentation training (Cohen et al., 2019) or the $\mathcal{L}_{\text{CRA}}$ loss proposed in our work.

**Datasets** We evaluate our approach on two academic datasets – `CIFAR-10` and `CIFAR-100` (Krizhevsky et al., 2009), and two commercial datasets – Mapillary Traffic Sign Dataset (`MTSD`) (Ertler et al., 2020) and a Rail Defect Dataset kindly provided by Swiss Federal Railways (`SBB`). We provide full details, including example images and all preprocessings steps, in the Appendix A.1.

When training on the `CIFAR-10` and `CIFAR-100` datasets, we use the AutoAugment (`AA`) policy by Cubuk et al. (2018) as the image augmentation. For the `MTSD` and `SBB` datasets, we use standard image augmentations (`SA`) consisting of random cropping, color jitter, and random translation and rotation. For completeness, our evaluation also includes models trained without any data augmentations.

---

[2]Our method can also be used to train from scratch, in which case a scheduler for $\beta$ should be introduced.

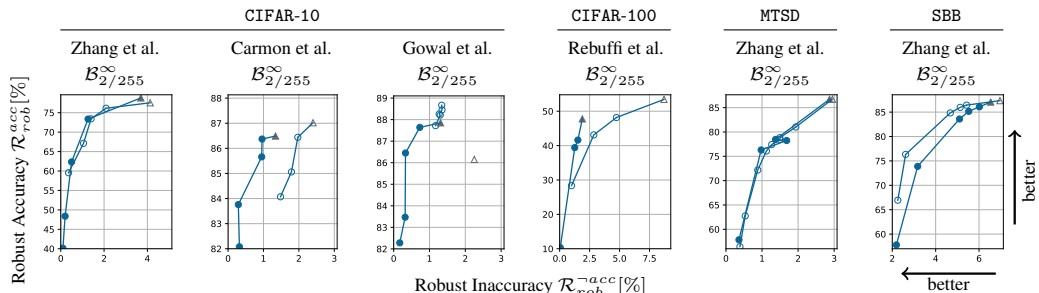

Figure 2: Robust accuracy ($\mathcal{R}_{rob}^{acc}$) and robust inaccuracy ($\mathcal{R}_{rob}^{\neg acc}$) of existing robust models (△, ▲) fine-tuned with our proposed loss (○, ●). Our approach consistently reduces the number of robust inaccurate samples across various datasets, existing models and at different regularization levels $\beta$.

**Metrics** We use the natural accuracy, robust accuracy, and robust inaccuracy as our main evaluation metrics, as defined in Section 3, but evaluated on the corresponding test dataset $\mathcal{D} = \{(\boldsymbol{x}_i, y_i)_{i=1}^N\}$.

When evaluating the empirical robustness, we use 40-step APGD (Croce & Hein, 2020) to evaluate the robustness of the classifier $F_\theta$. To evaluate certified robustness, we use the Monte Carlo algorithm for randomized smoothing from Cohen et al. (2019). We certify $500$ test samples and use the same randomized smoothing hyperparameters as Cohen et al. (2019) (cf. Appendix A.2).

## 7.1 REDUCING ROBUST INACCURACY

**Empirical Robustness** For empirically robust models, the results in Figure 2 show the robust accuracy ($\mathcal{R}_{rob}^{acc}$) and robust inaccuracy ($\mathcal{R}_{rob}^{\neg acc}$) of different existing models fine-tuned with (▲) and without (△) data augmentations. At the same time, Figure 2 also shows the same models fine-tuned with our proposed loss with (●) and without (○) data augmentations. We can see that our approach improves over the existing models across all the datasets. For example, on CIFAR-10 and for $\mathcal{B}_{2/255}^\infty$, the model from Carmon et al. (2019) achieves 86.5% robust accuracy, but also robust inaccuracy of 1.34%. In contrast, using our loss $\mathcal{L}_{\text{ERA}}$, we can obtain a number of models that reduce robust inaccuracy to 0.29%, while still achieving robustness of 83.8%. Similar results are obtained for other models, datasets, and perturbation regions (cf. Appendix A.5). We generally observe that our approach achieves consistently lower robust inaccuracy compared to adversarial training. Further, by varying the regularization term $\beta$, we obtain a Pareto front of optimal solutions.

**Certified Robustness** Similarly, we show the robust accuracy ($\mathcal{R}_{rob}^{acc}$) and robust inaccuracy ($\mathcal{R}_{rob}^{\neg acc}$) for certifiably robust models, which were fine-tuned using Gaussian noise augmentation $\mathcal{L}_{noise}$ and using our proposed loss function $\mathcal{L}_{\text{CRA}}$. In Table 2, we show results on CIFAR-10 for $\mathcal{B}_{0.12}^2$ and $\mathcal{B}_{0.25}^2$ perturbation regions. We can see that our approach achieves lower robust inaccuracy compared to existing models. For example, on CIFAR-10 and $\mathcal{B}_{0.25}^2$, the Cohen et al. (2019) model achieves 62% robust accuracy, but also 1% robust inaccuracy. In contrast, our approach reduces the robust inaccuracy to 0.4% while still achieving 53.8% robust accuracy. For the Sehwag et al. (2021) model, our approach even improves both the robust accuracy and the robust inaccuracy. For $\mathcal{B}_{0.25}^2$, our approach improves the robust accuracy by $+4.8\%$ and reduces the robust inaccuracy by $-0.6\%$.

| CIFAR-10 | | $\mathcal{B}_{0.12}^2$ ($\sigma = 0.06$) | | $\mathcal{B}_{0.25}^2$ ($\sigma = 0.12$) | |
|---|---|---|---|---|---|
| Pre-trained Model | Finetuning | $\mathcal{R}_{rob}^{acc}$ | $\mathcal{R}_{rob}^{\neg acc}$ | $\mathcal{R}_{rob}^{acc}$ | $\mathcal{R}_{rob}^{\neg acc}$ |
| Cohen et al. (2019) | $\mathcal{L}_{noise}$ | **74.0** | 2.8 | **62.0** | 1.0 |
| (ResNet-110) | $\mathcal{L}_{\text{CRA}}$ (ours) | 71.6 | **2.6** | 53.8 | **0.4** |
| Sehwag et al. (2021) | $\mathcal{L}_{noise}$ | 87.0 | 1.8 | 77.4 | 1.4 |
| (ResNet-18) | $\mathcal{L}_{\text{CRA}}$ (ours) | **90.8** | **1.6** | **82.2** | **0.8** |

Table 2: Robust accuracy ($\mathcal{R}_{rob}^{acc}$) and robust inaccuracy ($\mathcal{R}_{rob}^{\neg acc}$) of existing robust models fine-tuned with our proposed loss $\mathcal{L}_{\text{CRA}}$. All models are certified via randomized smoothing, using the hyperparameters listed in Appendix A.2.

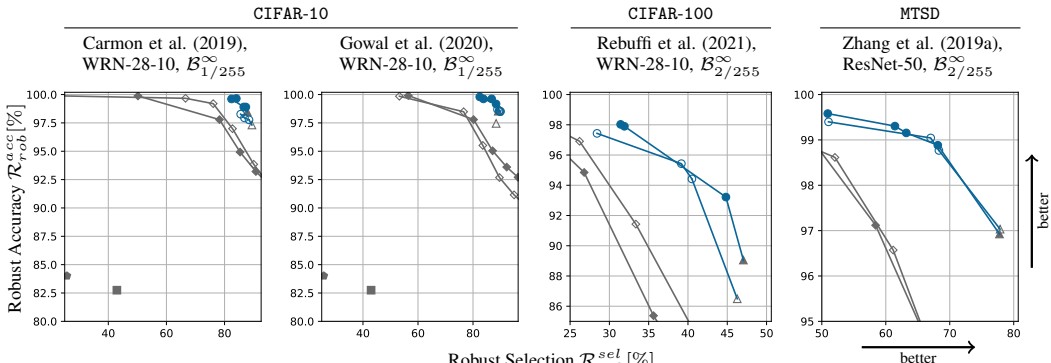

Figure 3: Comparison of different abstain approaches including existing robust classifiers $\text{TRADES}_{\text{RI}}$ ($\triangle$, $\blacktriangle$), classifiers fine-tuned with our proposed loss $\text{ERA}_{\text{RI}}$ ($\bigcirc$, $\bullet$), selection network ($\blacksquare$, $\pentagon$) and softmax response ($\Diamond$, $\blacklozenge$) abstain models. The higher $\mathcal{R}_{rob}^{sel}$ and $\mathcal{R}_{rob}^{acc}$, the better (top right corner is optimal).

## 7.2 USING ROBUSTNESS TO ABSTAIN

Next, we evaluate using robustness as an abstain mechanism (Section 5) and how it benefits from the training proposed in our work. We compare the following abstain mechanisms:

*Softmax Response (SR)* (Geifman & El-Yaniv, 2017), which abstains if the maximum softmax output of the model $f_\theta$ is below a threshold $\tau$ for some input $\boldsymbol{x}' \in \mathcal{B}_\varepsilon^p(\boldsymbol{x})$, that is:

$$S_{\text{SR}}(\boldsymbol{x}) = \mathbf{1}\{\forall \boldsymbol{x}' \in \mathcal{B}_\varepsilon^p(\boldsymbol{x}): \max_{c \in \mathcal{Y}} f_\theta(\boldsymbol{x}')_c \geq \tau\} \tag{16}$$

Similar to the robustness indicator (Section 5), to guarantee robustness of $S_{\text{SR}}$, we need to check the maximum softmax output of the model $f_\theta$ on double the region $\mathcal{B}_{2 \cdot \varepsilon}^p(\boldsymbol{x})$. To evaluate robustness of $S_{\text{SR}}$, we use a modified version of APGD called APGDconf (Appendix A.4). For each model considered in our work (e.g., Carmon et al. (2019)), we evaluate its corresponding abstain selector:

- ($\Diamond$, $\blacklozenge$) $\text{CARMON}_{\text{SR}}$, $\text{GOWAL}_{\text{SR}}$, $\text{ZHANG}_{\text{SR}}$, etc., all of which are fine-tuned using TRADES.

*Robustness Indicator (RI) (our work)*, which abstains if the model $F_\theta$ is non-robust:

$$S_{\text{RI}}(\boldsymbol{x}) = \mathbf{1}\{\forall \boldsymbol{x}' \in \mathcal{B}_\varepsilon^p(\boldsymbol{x}): F_\theta(\boldsymbol{x}') = F_\theta(\boldsymbol{x})\} \tag{17}$$

Note that, unlike other selectors, our robustness indicator is by design robust against an adversary using the same threat model. For each model, we compare two instantiations of this approach:

- ($\triangle$, $\blacktriangle$) $\text{TRADES}_{\text{RI}}$, ($\bigcirc$, $\bullet$) $\text{ERA}_{\text{RI}}$ (Equation 5).

*Selection Network (SN)*, which trains a separate neural network $s_\theta \colon \mathcal{X} \to \mathbb{R}$ and selects if:

$$S_{\text{SN}}(\boldsymbol{x}) = \mathbf{1}\{s_\theta(\boldsymbol{x}) \geq \tau\} \tag{18}$$

When considering the robustness of the abstain model, the robustness of both the classifier and the selection network have to be taken into account. We compare against two instantiations of this approach, both trained using certified training:

- ($\blacksquare$) $\text{ACE-COLT}_{\text{SN}}$ (Balunovic & Vechev, 2019; Mueller et al., 2020), and
- ($\pentagon$) $\text{ACE-IBP}_{\text{SN}}$ (Gowal et al., 2018; Mueller et al., 2020).

**Empirical Robustness** In Figure 3, we show the comparison of different abstain approaches using two metrics – the robust selection ($\mathcal{R}_{rob}^{sel}$), and the ratio of non-abstained samples that are robust and accurate ($\mathcal{R}_{rob}^{acc}$). Ideally, we would like both of these to be as high as possible, but typically there is a tradeoff between the two. This can clearly be seen in the results, where both our approach and softmax response can be used to obtain a Pareto front of optimal solutions.

Overall, the main results in Figure 3 show that as designed, our approach consistently improves robust accuracy. For example, on CIFAR-10 at $\varepsilon_\infty = 1/255$ and Carmon et al. (2019) model, we

| CIFAR-10 | | $\mathcal{B}^2_{0.12}$ ($\sigma = 0.06$) | | $\mathcal{B}^2_{0.25}$ ($\sigma = 0.12$) | |
|---|---|---|---|---|---|
| Pre-trained Model | Finetuning | $\mathcal{R}^{sel}_{rob}$ | $\mathcal{R}^{acc}_{rob}$ | $\mathcal{R}^{sel}_{rob}$ | $\mathcal{R}^{acc}_{rob}$ |
| Cohen et al. (2019) (ResNet-110) | $\mathcal{L}_{noise}$ | **76.8** | 96.35 | **63.0** | 98.41 |
| | $\mathcal{L}_{\text{CRA}}$ (ours) | 74.2 | **96.50** | 54.2 | **99.26** |
| Sehwag et al. (2021) (ResNet-18) | $\mathcal{L}_{noise}$ | 88.8 | 97.97 | 78.8 | 98.22 |
| | $\mathcal{L}_{\text{CRA}}$ (ours) | **92.4** | **98.27** | **83.0** | **99.04** |

Table 3: Comparison of our proposed loss $\mathcal{L}_{\text{CRA}}$ with the $\mathcal{L}_{noise}$ used in probabilistic certification. All models are certified via randomized smoothing, using the hyperparameters listed in Table A.2.

can successfully improve robust accuracy by +1.18% at the expense of -3.78% decrease in robust selection. This is close to optimal since increasing robust accuracy is typically achieved by correctly selecting for which samples to abstain. Interestingly, for some models and datasets, we can strictly improve over the baseline models by increasing both the robust accuracy and the non-abstained samples. On CIFAR-10 at $\varepsilon_\infty = 1/255$ and Gowal et al. (2020) model, we increase the robust accuracy by +1.06% and robust selection by +1.61% (when training without data augmentations).

Compared to the other abstain methods, our approach, in general, improves both metrics while also providing much stronger guarantees. Concretely, our approach guarantees that selected samples are robust in the considered threat model. Softmax response only guarantees that all samples in the considered threat model have high confidence and is thus vulnerable to high confidence adversarial examples, and the selection network provides no guarantees with regards to the selector's robustness.

**Certified Robustness** Applying our training for certified robustness $\mathcal{L}_{\text{CRA}}$ with $\beta = 1.0$ consistently improves robust accuracy $\mathcal{R}^{acc}_{rob}$ of robustness indicator abstain models. In Table 3, we show our results on CIFAR-10, using $\ell_2$ perturbations of radius 0.06 and 0.12. For instance, for the Cohen et al. (2019) model trained at $\sigma = 0.12$, we are able to improve the robust accuracy by +0.85% for $\varepsilon_2 = 0.25$ perturbations, at the expense of $-8.8\%$ decrease in robust selection. On the other hand, for the Sehwag et al. (2021) model, our approach improves on both metrics. For $\varepsilon_2 = 0.25$ perturbations, we increase robust accuracy by +0.82% and robust selection by +4.2%.

### 7.3 BOOSTING ROBUSTNESS WITHOUT ACCURACY LOSS

Finally, we present the results of combining the abstain models trained so far with state-of-the-art models trained to achieve high accuracy. Note that, as discussed in Section 5, when evaluating adversarial robustness for $\mathcal{B}^p_\varepsilon$, we in fact need to consider $\mathcal{B}^p_{2\cdot\varepsilon}$ robustness of the abstain model.

A summary of the results is shown in Figure 4. Similar to the results shown so far, the 2-compositional architectures that use models trained by our method (○, ●) improve over existing methods that optimize only for robust accuracy (△, ▲), as well as over models using softmax response (◇, ◆) or selection network (■, ⬠) to abstain. For example, for CIFAR-10 with $\varepsilon_\infty = 1/255$ and the Carmon et al. (2019) model, we improve natural accuracy by +0.58% and +0.62%, while decreasing the robustness only by -2.75% and -2.82%, when training with and without data augmentations respectively.

More importantly, our approach significantly improves robustness of highly accurate non-compositional models, with minimal loss of accuracy. In fact, in half of the cases, the compositional architecture even slightly improves the overall accuracy, as summarized in Table 4. We provide full results, including additional models and perturbation bounds in Appendix A.7, and an evaluation of the considered highly accurate non-compositional models in Appendix A.8.

| | CIFAR-10 | CIFAR-100 | MTSD | SBB |
|---|---|---|---|---|
| | (Zhao et al. (2020)), $\mathcal{B}^\infty_{1/255}$ | (WideResNet-28-10), $\mathcal{B}^\infty_{2/255}$ | (ResNet-50), $\mathcal{B}^\infty_{2/255}$ | (ResNet-50), $\mathcal{B}^\infty_{2/255}$ |
| $\mathcal{R}^{acc}_{rob}$ | 26.2 $\xrightarrow{+\mathbf{60.3}\%}$ 86.5 | 3.1 $\xrightarrow{+\mathbf{38.8}\%}$ 41.9 | 40.7 $\xrightarrow{+\mathbf{29.2}\%}$ 69.9 | 44.7 $\xrightarrow{+\mathbf{37.7}\%}$ 82.4 |
| $\mathcal{R}_{nat}$ | 97.8 $\xrightarrow{-0.2\%}$ 97.6 | 80.17 $\xrightarrow{+\mathbf{0.01}\%}$ 80.18 | 93.8 $\xrightarrow{+\mathbf{0.2}\%}$ 94.0 | 91.4 $\xrightarrow{-0.1\%}$ 91.3 |

Table 4: Improvement of applying our approach to models trained to optimize natural accuracy only.

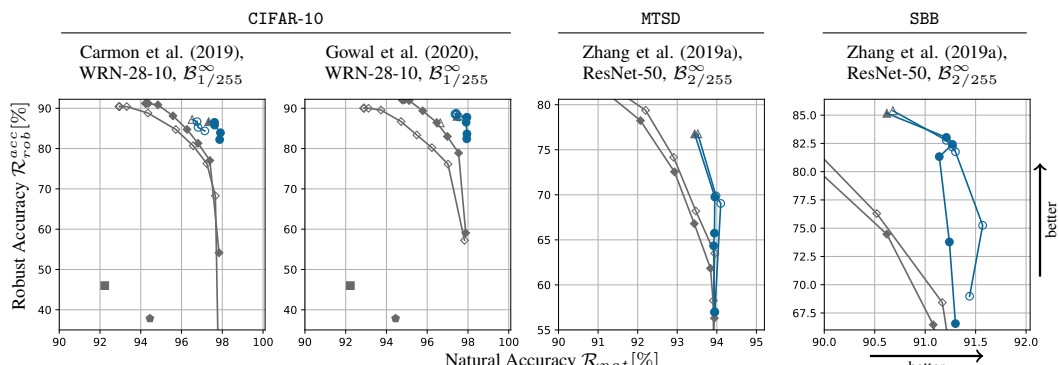

Figure 4: Natural ($\mathcal{R}_{nat}$) and robust accuracy ($\mathcal{R}_{rob}^{acc}$) for 2-compositional ERA$_\text{RI}$ models (○, ●) and 2-compositional TRADES$_\text{RI}$ models (△, ▲). Further, we also consider 2-compositional ACE-COLT$_\text{SN}$, ACE-IBP$_\text{SN}$ (■, ⬟), and 2-compositional TRADES$_\text{SR}$ (◇, ◆) models. The core models used in the compositional architectures are listed in Appendix A.8.

## 8 CONCLUSION

In this work, we address the issue of robust inaccuracy of state-of-the-art robust models. Ideally, models should be robust only if accurate, however, we show that existing robust models have non-negligible amounts of robust but inaccurate samples. We present a new training method that jointly minimizes robust inaccuracy and maximizes robust accuracy. The key concept was extending an existing robust training loss with a term that minimizes robust inaccuracy, making our method widely applicable since it can be instantiated using various existing robust training methods. We show the practical benefits of our approach by both, using robustness as an abstain mechanism, and by leveraging compositional architectures to improve robustness without sacrificing accuracy.

However, there are also limitations and interesting extensions to consider in future work. First, while there are some cases where our training improves both robust accuracy and reduces robust inaccuracy (e.g., for certified robust Sehwag et al. (2021) model on `CIFAR-10` or empirically robust Gowal et al. model on `CIFAR-10`), it does typically results in a trade-off between the two – reduced robust inaccuracy also leads to reduced robust accuracy. To address this issue, in practice we compute a Pareto front of optimal solutions, all of which can be used to instantiate the compositional model. An interesting future work would be to explore this trade-off further and develop new techniques to mitigate it. Second, given that we compute a Pareto front of the optimal solutions, a promising future work item is considering model cascades that consist of different models along this Pareto front, and progressively fall back to models with higher robust accuracy but also higher robust inaccuracy. Third, we observed that as the robust inaccuracy approaches zero (i.e., the best case), the training becomes much harder. This is both because these remaining robust inaccurate examples are the hardest to fix, as well as because there are only very few of them. In our work, we explored using data augmentation to address this issue, but more work is needed to make the training efficient in such a low data regime.

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

# A APPENDIX

## A.1 DATASETS

We ran our evaluations on four different datasets, namely on `CIFAR-10` and `CIFAR-100` (Krizhevsky et al., 2009), the Mapillary Traffic Sign Dataset (`MTSD`) (Ertler et al., 2020), and a rail defect dataset provided by Swiss Federal Railways (`SBB`). Additionally, we used a synthetic dataset consisting of two-dimensional data points. In the following, we explain the necessary preprocessing steps to create the publicly available `MTSD` dataset.

**Mapillary Traffic Sign Dataset (MTSD)** The Mapillary traffic sign dataset (Ertler et al., 2020) is a large-scale vision dataset that includes 52'000 fully annotated street-level images from all around the world. The dataset covers 400 known and other unknown traffic signs, resulting in over 255'000 traffic signs in total. Each street-level image is manually annotated and includes ground truth bounding boxes that locate each traffic sign in the image, as shown in Figure 5a. Further, each ground truth traffic sign annotation includes additional attributes such as ambiguousness or occlusion. Since the focus of this work is on classification, we convert the base `MTSD` dataset to a classification dataset (described below) by cropping to each ground truth bounding box. We show samples from the resulting cropped `MTSD` dataset in Figure 5b.

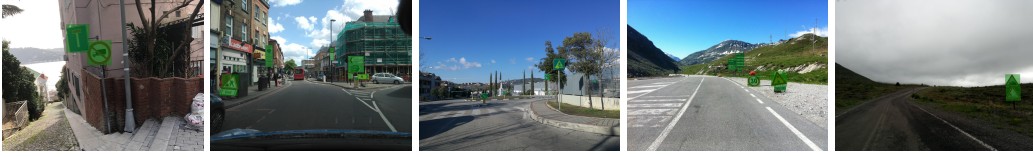

(a) Base Mapillary Traffic Sign Dataset (`MTSD`). The ground truth bounding boxes are visualized in green.

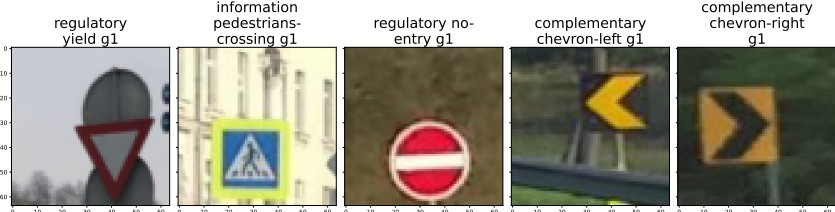

(b) Preprocessed Mapillary Traffic Sign Dataset (`MTSD`).

Figure 5: Illustration of Mapillary Traffic Sign Dataset (`MTSD`) samples. The base dataset consists of street-level images that include annotated ground truth bounding boxes locating the traffic signs (a). We convert the dataset to a classification task by cropping to the ground truth bounding boxes (b).

We convert the `MTSD` objection detection dataset into a classification dataset as follows:

1. Ignore all bounding boxes that are annotated as occluded (sign partly occluded), out-of-frame (sign cut off by image border), exterior (sign includes other signs), ambiguous (sign is not classifiable at all), included (sign is part of another bigger sign), dummy (looks like a sign but is not) (Ertler et al., 2020). Further, we ignore signs of class *other-sign*, since this is a general class that includes any traffic sign with a label not within the `MTSD` taxonomy.

2. Crop to all remaining bounding boxes and produce a labeled image classification dataset. Cropping is done with slack, i.e. we crop to a randomly upsized version of the original bounding box. Given a bounding box $BB = ([x_{min}, x_{max}], [y_{min}, y_{max}])$, the corresponding upsized bounding box is given as

$$
\begin{aligned}
UBB = \big( & [x_{min} - \lambda\alpha_x(x_{max} - x_{min}),\ x_{max} + \lambda(1 - \alpha_x)(x_{max} - x_{min})], \\
& [y_{min} - \lambda\alpha_y(y_{max} - y_{min}),\ y_{max} + \lambda(1 - \alpha_y)(y_{max} - y_{min})] \big)
\end{aligned}
\tag{19}
$$

where $\alpha_x, \alpha_y \sim \mathcal{U}_{[0,1]}$ [3] and $\lambda$ is the slack parameter, which we set to $\lambda = 1.0$.

3. Resize cropped traffic signs to $(64, 64)$.

[3] $\mathcal{U}_{[a,b]}$ is the uniform distribution over the interval $[a, b]$.

**Rail Defect Dataset (SBB)** The rail defect dataset (SBB) is a proprietary vision dataset collected and annotated by Swiss Federal Railways. It includes images of rails, each of which is annotated with ground truth bounding boxes for various types of rail defects. We note that all the models used in our work for this dataset are trained by the authors and not provided by SBB. In fact, for our work, we even consider a different type of task – classification instead of the original object detection. As a consequence, the accuracy and robustness results presented in our work are by no means representative of the actual models used by SBB.

## A.2   HYPERPARAMETERS

**TRADES** We use $\mathcal{L}_{\text{TRADES}}$ (Zhang et al., 2019a) to both train models from scratch and fine-tune existing models. When training models from scratch, we train for 100 epochs using $\mathcal{L}_{\text{TRADES}}$, with an initial learning rate 1e-1, which we reduce to 1e-2 and 1e-3, once 75% and 90% of the total epochs are completed. When fine-tuning models, we train for 50 epochs using $\mathcal{L}_{\text{TRADES}}$, with an initial learning rate 1e-3, which we reduce to 1e-4 once 75% of the total epochs are completed. We use batch size 200, use 10-step PGD (Madry et al., 2018) to generate adversarial examples during training, and set the $\beta$ parameter in $\mathcal{L}_{\text{TRADES}}$ to $\beta_{TRADES} = 6.0$.

**Empirical Robustness Abstain Training** We fine-tune for 50 epochs using $\mathcal{L}_{\text{ERA}}$ (Equation 5), with an initial learning rate 1e-3, which we reduce to 1e-4 once 75% of the total epochs are completed. We use batch size 200, use 10-step PGD (Madry et al., 2018) to generate adversarial examples during training, and set $\beta_{TRADES} = 6.0$ for the loss term $\mathcal{L}_{rob} = \mathcal{L}_{\text{TRADES}}$.

**Certified Robustness Abstain Training** We fine-tune for 50 epochs using $\mathcal{L}_{\text{CRA}}$ (Equation 9), with an initial learning rate 1e-3, which we reduce to 1e-4 once 75% of the total epochs are completed. We use batch size 50, $k = 16$ *i.i.d.* samples from $\mathcal{N}(0, \sigma^2 \boldsymbol{I})$, and set the inverse softmax temperature to $\Gamma = 4.0$ (cf. Section 4.2).

**Probabilistic Certification via Randomized Smoothing** We use the practical Monte Carlo algorithm by Cohen et al. (2019) for randomized smoothing, using the same certification hyperparameters as them. We use $N_0 = 100$ Monte Carlo samples to identify the most probable class $c_A$, $N = 100,000$ Monte Carlo samples to estimate a lower bound on the probability $p_A$, and set the failure probability to $\alpha = 0.001$.

**Synthetic Dataset** In Figure 1, we illustrate the effect of our training on a synthetic three-class dataset, where each class follows a Gaussian distribution. We then use a simple four-layer neural network with 64 neurons per layer, and train it on $N = 1000$ synthetic samples, using $\mathcal{L}_{std}$, $\mathcal{L}_{\text{TRADES}}$ (Zhang et al., 2019a), and $\mathcal{L}_{\text{ERA}}$ (Equation 5). For each loss variant, we train for 20 epochs, use a fixed learning rate 1e-1, and batch size 10. For $\mathcal{L}_{\text{TRADES}}$ and $\mathcal{L}_{\text{ERA}}$, we use 10-step PGD (Madry et al., 2018) to generate adversarial examples during training, and set $\beta_{TRADES} = 6.0$.

## A.3   LOSS FUNCTION ABLATION STUDY

Additionally to the $\mathcal{L}_{\text{ERA}}$ loss from Equation 5, we consider an alternative loss formulation for training an empirical robustness indicator abstain model. The formulation is based on the Deep Gamblers loss (Liu et al., 2019), which considers an abstain model $(F_\theta, S)$ with an explicit abstain class $a$ as a selection mechanism. Since we consider robustness indicator selection, we replace the output probability of the abstain class $f_\theta(\boldsymbol{x})_a$ with the output probability of the most likely adversarial label. This corresponds to the probability of a sample being non-robust and thus the probability of abstaining under a robustness indicator selector. Similar to $\mathcal{L}_{\text{ERA}}$, we also add the TRADES loss (Zhang et al., 2019a) to optimize robust accuracy. The resulting loss is then defined as:

$$\mathcal{L}_{\text{DGA}}(f_\theta, (\boldsymbol{x}, y)) = \beta \cdot \mathcal{L}_{\text{TRADES}}(f_\theta, (\boldsymbol{x}, y)) - \log\left(f_\theta(\boldsymbol{x})_y + \max_{c \in \mathcal{Y} \setminus \{F_\theta(\boldsymbol{x})\}} f_\theta(\boldsymbol{x}')_c\right) \quad (20)$$

We conduct an ablation study over the two loss functions, $\mathcal{L}_{\text{ERA}}$ and $\mathcal{L}_{\text{DGA}}$, for CIFAR-10 and a $\varepsilon_\infty = 8/255$ TRADES (Zhang et al., 2019a) trained ResNet-50 model. We fine-tune the model for $\ell_\infty$ perturbations of radii $1/255$ and $2/255$, using both $\mathcal{L}_{\text{ERA}}$ and $\mathcal{L}_{\text{DGA}}$, training for 50 epochs each and setting the regularization parameter $\beta = 1.0$. For each loss variant, we train the base model once without data augmentations and once using the AutoAugment (AA) policy (Cubuk et al., 2018).

| CIFAR-10 | | $\mathcal{B}^\infty_{1/255}$ | | $\mathcal{B}^\infty_{2/255}$ | |
|---|---|---|---|---|---|
| Pre-trained Model | Finetuning | $\mathcal{R}^{sel}_{rob}$ | $\mathcal{R}^{acc}_{rob}$ | $\mathcal{R}^{sel}_{rob}$ | $\mathcal{R}^{acc}_{rob}$ |
| | $\mathcal{L}_{\text{ERA}}$ | **86.31** | **96.63** | **78.24** | **97.33** |
| Zhang et al. (2019a) | $\mathcal{L}_{\text{DGA}}$ | 84.98 | 94.92 | 75.73 | 96.22 |
| (ResNet-50) | $\mathcal{L}_{\text{ERA}} + AA$ | **83.44** | **97.47** | **74.63** | **98.31** |
| | $\mathcal{L}_{\text{DGA}} + AA$ | 80.72 | 96.56 | 73.59 | 97.88 |

Table 5: Robust selection ($\mathcal{R}^{sel}_{rob}$) and robust accuracy ($\mathcal{R}^{acc}_{rob}$) of empirical robustness indicator abstain models ($F, S_{\text{ERI}}$), trained using $\mathcal{L}_{\text{ERA}}$ (Equation 5) and $\mathcal{L}_{\text{DGA}}$ (Equation 20).

We show the robust accuracy and the robust selection of the resulting robustness indicator abstain models in Table 5. Observe that for all experiments, $\mathcal{L}_{\text{ERA}}$ trained models achieve consistently higher robust accuracy and higher robust selection, compared to $\mathcal{L}_{\text{DGA}}$ trained models. For instance, when training for $\varepsilon_\infty = 1/255$ perturbations without data augmentations, $\mathcal{L}_{\text{ERA}}$ achieves $+1.71\%$ higher robust accuracy and $+1.33\%$ higher robust selection, compared to $\mathcal{L}_{\text{DGA}}$. Similarly, when training with AutoAugment, $\mathcal{L}_{\text{ERA}}$ achieves $+0.91\%$ higher robust accuracy and $+2.72\%$ higher robust selection. Similar results hold for $\varepsilon_\infty = 2/255$ perturbations.

## A.4 Comparing Adversaries for Softmax Response (SR)

Recall from Section 7.2 that we evaluated the robustness of softmax response (SR) abstain models using `APGDconf`, which is a modified version of APGD (Croce & Hein, 2020) using the alternative adversarial attack objective by Stutz et al. (2020). This modified objective optimizes for an adversarial example $\boldsymbol{x}'$ that maximizes the confidence in any label $c \neq F_\theta(\boldsymbol{x})$, instead of minimizing the confidence in the predicted label:

$$\boldsymbol{x}' = \arg\max_{\hat{\boldsymbol{x}} \in \mathcal{B}^p_\varepsilon(\boldsymbol{x})} \max_{c \neq F_\theta(\boldsymbol{x})} f_\theta(\hat{\boldsymbol{x}})_c \tag{21}$$

The resulting adversarial attack finds high confidence adversarial examples, and thus represents an effective attack against a softmax response selector $S_{\text{SR}}$.

In the following, we conduct an ablation study over `APGD` and `APGDconf` by evaluating the robust selection $\mathcal{R}^{sel}_{rob}$ and robust accuracy $\mathcal{R}^{acc}_{rob}$ of an SR abstain model ($F_\theta, S_{\text{SR}}$) using both `APGD` and `APGDconf`. We use the adversarially trained WideResNet-28-10 model by Carmon et al. (2019) (taken from RobustBench (Croce et al., 2020)), trained on `CIFAR-10` for $\varepsilon_\infty = 8/255$ perturbations. We then evaluate the classifier as an SR abstain model ($F_\theta, S_{\text{SR}}$) with varying threshold $\tau \in [0, 1)$, and report the robust selection and robust accuracy for varying $\ell_\infty$ perturbations in Figure 6. Observe that for small perturbations such as $\varepsilon_\infty = 1/255$, `APGD` and `APGDconf` are mostly equivalent concerning robust selection and robust accuracy. However, for larger perturbations such as $\varepsilon_\infty = 4/255$, the SR abstain model is significantly less robust to `APGDconf` than to standard `APGD`, showing the importance of choosing a suitable adversarial attack. High confidence adversarial examples are generally more likely to be found for larger perturbations, thus an SR selector is significantly less robust to `APGDconf` than to `APGD` for larger perturbations.

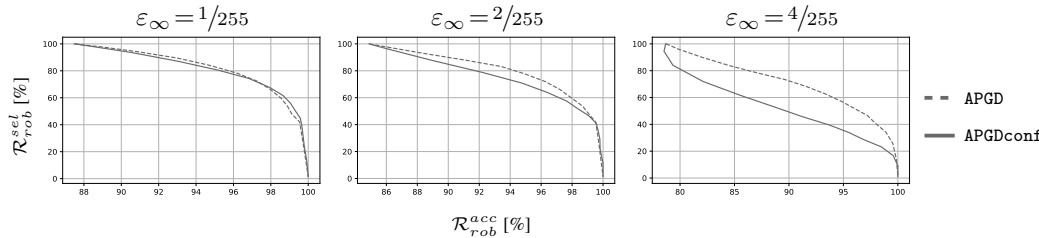

Figure 6: Robust selection ($\mathcal{R}^{sel}_{rob}$) and robust accuracy ($\mathcal{R}^{acc}_{rob}$) for `CIFAR-10` softmax response (SR) abstain models ($F, S_{\text{SR}}$), for varying threshold $\tau \in [0, 1)$ and using the WideResNet-28-10 classifier $F$ by Carmon et al. (2019). Each SR abstain model is evaluated via `APGD` (Croce & Hein, 2020) and `APGDconf` (Equation 21).

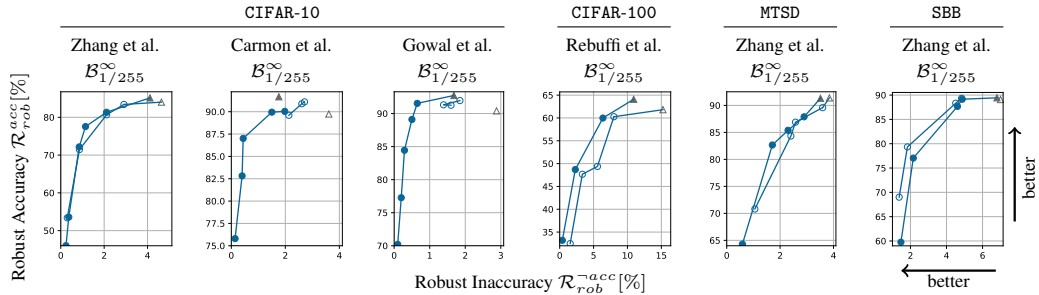

Figure 7: Robust accuracy ($\mathcal{R}_{rob}^{acc}$) and robust inaccuracy ($\mathcal{R}_{rob}^{\neg acc}$) of existing robust models ($\triangle$, $\blacktriangle$) fine-tuned with our proposed loss ($\circ$, $\bullet$). Our approach consistently reduces the number of robust inaccurate samples across various datasets, existing models and at different regularization levels $\beta$.

## A.5 Additional Experiments on Reducing Robust Inaccuracy

In this section, we present additional experiments on reducing robust inaccuracy for empirical robustness.

Similar to the results in Figure 2, we show the robust accuracy ($\mathcal{R}_{rob}^{acc}$) and robust inaccuracy ($\mathcal{R}_{rob}^{\neg acc}$) of different existing models fine-tuned with ($\blacktriangle$) and without ($\triangle$) data augmentations, in Figure 7. At the same time, Figure 7 also shows the same models fine-tuned with our proposed loss with ($\bullet$) and without ($\circ$) data augmentations. We again observe that our approach achieves consistently lower robust robust inaccuracy, compared to existing robust models. For example, on CIFAR-10 and for $\mathcal{B}_{1/255}^{\infty}$, the model from Carmon et al. (2019) achieves $91.7\%$ robust accuracy but also $1.8\%$ robust inaccuracy. Using our loss $\mathcal{L}_{ERA}$ and varying the regularization term $\beta$, we can obtain a number of models that reduce robust inaccuracy to $0.14\%$ while still achieving robust accuracy of $75.8\%$.

## A.6 Additional Experiments on Using Robustness to Abstain

In this section, we present additional experiments on comparing different abstain approaches for empirical robustness.

We compare robustness indicator abstain models $(F, S_{RI})$ using existing robust classifiers TRADES$_{RI}$ and classifiers fine-tuned with our proposed loss ERA$_{RI}$. Further, we again consider softmax response and selection network abstain models, as described in Section 7.2. Equivalent to Section 7.2, we use the robust selection ($\mathcal{R}_{rob}^{sel}$), and the ratio of non-abstained samples that are robust and accurate ($\mathcal{R}_{rob}^{acc}$) as our evaluation metrics.

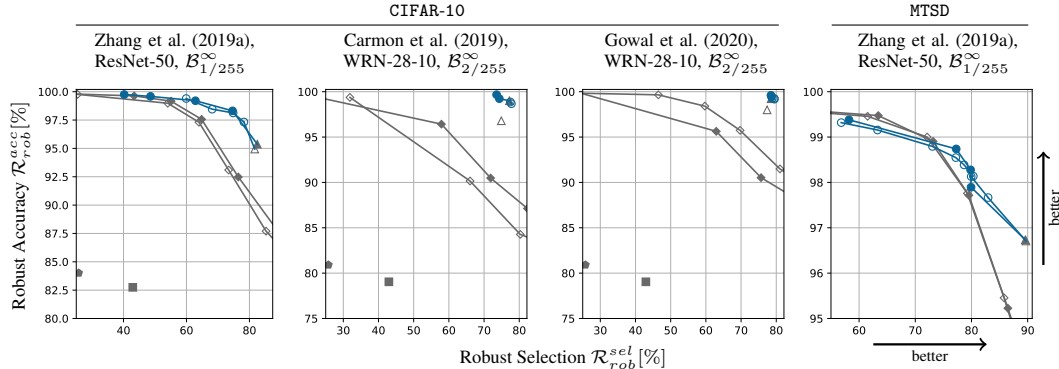

Figure 8: Comparison of different abstain approaches including existing robust classifiers TRADES$_{RI}$ ($\triangle$, $\blacktriangle$), classifiers fine-tuned with our proposed loss ERA$_{RI}$ ($\circ$, $\bullet$), selection network ($\blacksquare$, $\pentagon$) and softmax response ($\diamond$, $\blacklozenge$) abstain models. The higher $\mathcal{R}_{rob}^{sel}$ and $\mathcal{R}_{rob}^{acc}$, the better (top right corner is optimal).

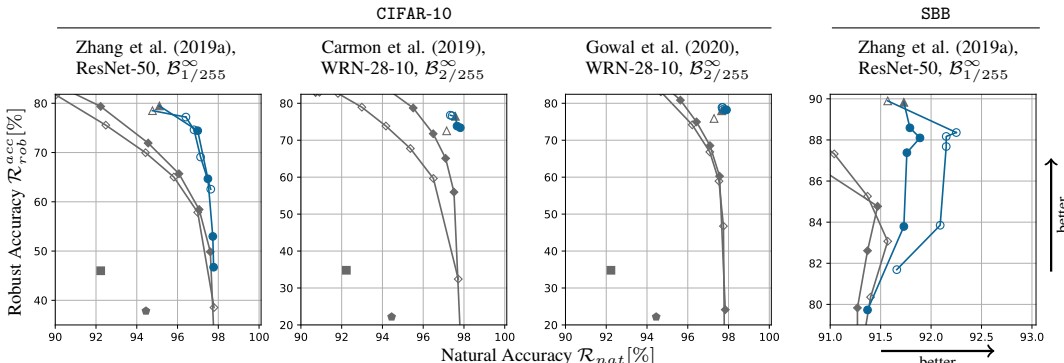

Figure 9: Natural ($\mathcal{R}_{nat}$) and robust accuracy ($\mathcal{R}_{rob}^{acc}$) for 2-compositional ERA$_{\text{RI}}$ models (O, ●) and 2-compositional TRADES$_{\text{RI}}$ models (△, ▲). Further, we also consider 2-compositional ACE-COLT$_{\text{SN}}$, ACE-IBP$_{\text{SN}}$ (■, ⬠), and 2-compositional TRADES$_{\text{SR}}$ (◇, ◆) models. The core models used in the compositional architectures are listed in Appendix A.8.

We show the comparison of the different abstain models in Figure 8. Similar to the results in Section 7.2, we again show that, as designed, our approach consistently improves robust accuracy. For instance, consider the CIFAR-10 Zhang et al. (2019a) model at $\varepsilon_\infty = {}^{1}/_{255}$, trained without data augmentations (O). The ERA$_{\text{RI}}$ model with the highest robust selection $\mathcal{R}_{rob}^{sel}$ improves robust accuracy by +2.39% at the expense of -3.44% decrease in robust selection. This tradeoff is close to optimal since our approach increases robust accuracy by correctly abstaining from mispredicted samples, thus an increase in robust accuracy results in a corresponding decrease in robust selection. Further, we again observe that by varying the regularization parameter $\beta$, we can obtain a Pareto front of optimal solutions. Considering the CIFAR-10 Zhang et al. (2019a) model at $\varepsilon_\infty = {}^{1}/_{255}$, trained with data augmentations (●), we can improve the robust accuracy up to 99.75%, an increase of +4.38% compared to the corresponding TRADES$_{\text{RI}}$ model (▲). However, this comes at the expense of a disproportionally large decrease of -42.27% lower robust selection. We observe similar results for other models, datasets, and perturbations regions, shown in Figure 8.

Further, we again note that our approach mostly improves both robust selection and robust accuracy when compared to softmax response and selection network abstain models.

## A.7 ADDITIONAL EXPERIMENTS ON BOOSTING ROBUSTNESS WITHOUT ACCURACY LOSS

In this section, we present additional results on combining abstain models with state-of-the-art models trained to achieve high natural accuracy.

Equivalent to Section 7.3, we put the abstain models trained so far in 2-composition (Section 6) with the standard trained core models discussed in Appendix A.8. We show the natural ($\mathcal{R}_{nat}$) and adversarial accuracy ($\mathcal{R}_{rob}^{acc}$) of the resulting 2-compositional architectures in Figure 9.

We again observe that 2-compositional architectures using models trained by our method (O, ●) improve over existing methods that solely optimize for robust accuracy (△, ▲). Further, our method mostly improves both the natural and robust accuracy, compared to 2-compositional architectures using softmax response (◇, ◆) or selection network (■, ⬠) to abstain. For example, on SBB and the Zhang et al. (2019a) model at $\varepsilon_\infty = {}^{1}/_{255}$, our approach (O) improves natural accuracy by +0.68%, while decreasing the robust accuracy by only -1.54%.

Further, we show that 2-compositional architectures using models trained by our method achieve significantly higher robustness and mostly equivalent overall accuracy, compared to state-of-the-art non-compositional models trained for high natural accuracy. In Table 6, we show the natural ($\mathcal{R}_{nat}$) and adversarial accuracy ($\mathcal{R}_{rob}^{acc}$) of our 2-compositional models and illustrate the accuracy improvement over the standard trained models discussed in Appendix A.8. For instance, consider CIFAR-10 at $\varepsilon_\infty = {}^{2}/_{255}$ and the 2-compositional architecture using the Gowal et al. (2020) model as robust model $F_{robust}$. Our model improves the robust accuracy by +75.3% and the natural accuracy

by +0.1%, compared to the standard trained model by Zhao et al. (2020). Similar results hold for other models, datasets, and perturbation regions.

| | | CIFAR-10 | | CIFAR-100 | MTSD | SBB |
|---|---|---|---|---|---|---|
| | $F_{core}$ | (Zhao et al., 2020) | | (WideResNet-28-10) | (ResNet-50) | (ResNet-50) |
| | $F_{robust}$ | Carmon et al. | Gowal et al. | Rebuffi et al. | Zhang et al. | Zhang et al. |
| $\mathcal{B}_{1/255}^{\infty}$ | $\mathcal{R}_{rob}^{acc}$ | 86.5 (+60.3%) | 87.8 (+61.6%) | 44.0 (+24.1%) | 84.5 (+9.8%) | 88.4 (+12.7%) |
| | $\mathcal{R}_{nat}$ | 97.6 (-0.2%) | 98.0 (+0.2%) | 80.5 (+0.3%) | 94.1 (+0.3%) | 92.3 (+0.9%) |
| $\mathcal{B}_{2/255}^{\infty}$ | $\mathcal{R}_{rob}^{acc}$ | 73.4 (+70.5%) | 78.2 (+75.3%) | 41.9 (+38.8%) | 69.9 (+29.2%) | 82.4 (+37.7%) |
| | $\mathcal{R}_{nat}$ | 97.8 (+0.0%) | 97.9 (+0.1%) | 80.18 (+0.01%) | 94.0 (+0.2%) | 91.3 (-0.1%) |

Table 6: Improvements of 2-compositional architectures using models $F_{robust}$ trained with our method over non-compositional models trained to optimize natural accuracy only (Appendix A.8).

## A.8  CORE MODELS

Recall from Section 6 that an abstain model $(F, S)$ can be enhanced by a core model $F_{core}$, which makes a prediction on all abstained samples, resulting in 2-compositional architectures. In Section 7.3, we presented an evaluation of 2-compositional architectures, where we used state-of-the-art standard trained models as core models. In Table 7, we show the natural and adversarial accuracy of core models used in Section 7.3, for varying $\ell_{\infty}$ perturbation regions, where we use 40-step APGD (Croce & Hein, 2020) to evaluate robustness.

| Dataset | Model $F_{core}$ | $\mathcal{R}_{nat}$ [%] | $\mathcal{R}_{rob}^{acc}$ [%] | | |
|---|---|---|---|---|---|
| | | | $\mathcal{B}_{1/255}^{\infty}$ | $\mathcal{B}_{2/255}^{\infty}$ | $\mathcal{B}_{4/255}^{\infty}$ |
| CIFAR-10 | Zhao et al. (2020) (WideResNet-40-10) | 97.81 | 26.18 | 2.92 | 0.06 |
| CIFAR-100 | (WideResNet-28-10) | 80.17 | 19.9 | 3.06 | 0.15 |
| MTSD | (ResNet-50) | 93.79 | 74.66 | 40.71 | 7.51 |
| SBB | (ResNet-50) | 91.37 | 75.65 | 44.69 | 8.76 |

Table 7: Natural ($\mathcal{R}_{nat}$) and adversarial accuracy ($\mathcal{R}_{rob}^{acc}$) of standard trained core models, used in 2-compositional architectures in Section 7.3 and Appendix A.7.

## A.9  ROBUSTNESS/ACCURACY DATASET SPLITS

Consider a robustness indicator abstain model $(F_\theta, S_{\text{RI}})$ and a labeled dataset $D = \{(\boldsymbol{x}_i, y_i)_{i=1}^{N}\}$ on which we evaluate the classifier $F_\theta \colon \mathcal{X} \to \mathcal{Y}$. Based on the robustness and accuracy of the classifier $F_\theta$, we can partition $D$ into four disjoint subsets $D = \{D_{F_\theta}^{r \wedge a}, D_{F_\theta}^{\neg r \wedge a}, D_{F_\theta}^{r \wedge \neg a}, D_{F_\theta}^{\neg r \wedge \neg a}\}$, where:

$$D_{F_\theta}^{r \wedge a} = \{(\boldsymbol{x}, y) \in D \colon \forall \boldsymbol{x}' \in \mathcal{B}_\varepsilon^p(\boldsymbol{x}). \, F_\theta(\boldsymbol{x}') = F_\theta(\boldsymbol{x}) \wedge F_\theta(\boldsymbol{x}) = y\}$$
$$D_{F_\theta}^{r \wedge \neg a} = \{(\boldsymbol{x}, y) \in D \colon \forall \boldsymbol{x}' \in \mathcal{B}_\varepsilon^p(\boldsymbol{x}). \, F_\theta(\boldsymbol{x}') = F_\theta(\boldsymbol{x}) \wedge F_\theta(\boldsymbol{x}) \neq y\}$$
$$D_{F_\theta}^{\neg r \wedge a} = \{(\boldsymbol{x}, y) \in D \colon \exists \boldsymbol{x}' \in \mathcal{B}_\varepsilon^p(\boldsymbol{x}). \, F_\theta(\boldsymbol{x}') \neq F_\theta(\boldsymbol{x}) \wedge F_\theta(\boldsymbol{x}) = y\}$$
$$D_{F_\theta}^{\neg r \wedge \neg a} = \{(\boldsymbol{x}, y) \in D \colon \exists \boldsymbol{x}' \in \mathcal{B}_\varepsilon^p(\boldsymbol{x}). \, F_\theta(\boldsymbol{x}') \neq F_\theta(\boldsymbol{x}) \wedge F_\theta(\boldsymbol{x}) \neq y\}$$

We illustrate this dataset partitioning on the CIFAR-10 (Krizhevsky et al., 2009) dataset. We consider a TRADES (Zhang et al., 2019b) trained ResNet-50 and the WideResNet-28-10 models by Carmon et al. (2019); Gowal et al. (2020) (taken from Robustbench (Croce et al., 2020)), where each model is adversarially pretrained for $\varepsilon_\infty = {}^{8}/_{255}$ and then fine-tuned via TRADES to the respective $\ell_\infty$ threat model illustrated Table 8. Further, we also consider a standard trained ResNet-50. We then evaluate the robustness and accuracy of each model using 40-step APGD (Croce & Hein, 2020). Considering Table 8, note that standard adversarial training methods do not necessarily eliminate the occurrence of robust inaccurate samples $(\boldsymbol{x}, y) \in D_{F_\theta}^{r \wedge \neg a}$, and that the robust inaccuracy generally increases for smaller perturbation regions. Further, we note that while standard trained models have low robust inaccuracy, they also have low overall robustness, resulting in low overall robust accuracy.

| Threat Model | Data Split | Relative Split Size [%] | | | |
|---|---|---|---|---|---|
| | | Zhang et al. (ResNet-50) | Carmon et al. (WRN-28-10) | Gowal et al. (WRN-28-10) | $\mathcal{L}_{std}$ (ResNet-50) |
| $\mathcal{B}_{1/255}^\infty$ | $\blacksquare|D_{F_\theta}^{\neg r \wedge \neg a}|$ | 5.17 | 3.33 | 2.85 | 6.97 |
| | $\blacksquare|D_{F_\theta}^{r \wedge \neg a}|$ | 4.64 | 3.61 | 2.88 | 0.0 |
| | $\blacksquare|D_{F_\theta}^{\neg r \wedge a}|$ | 6.18 | 3.32 | 3.87 | 74.89 |
| | $\blacksquare|D_{F_\theta}^{r \wedge a}|$ | 84.01 | 89.74 | 90.40 | 18.14 |
| $\mathcal{B}_{2/255}^\infty$ | $\blacksquare|D_{F_\theta}^{\neg r \wedge \neg a}|$ | 7.94 | 7.38 | 4.86 | 6.97 |
| | $\blacksquare|D_{F_\theta}^{r \wedge \neg a}|$ | 4.13 | 2.40 | 2.25 | 0.0 |
| | $\blacksquare|D_{F_\theta}^{\neg r \wedge a}|$ | 10.38 | 3.20 | 6.74 | 91.80 |
| | $\blacksquare|D_{F_\theta}^{r \wedge a}|$ | 77.55 | 87.02 | 86.15 | 1.23 |
| $\mathcal{B}_{4/255}^\infty$ | $\blacksquare|D_{F_\theta}^{\neg r \wedge \neg a}|$ | 13.42 | 8.23 | 6.64 | 6.97 |
| | $\blacksquare|D_{F_\theta}^{r \wedge \neg a}|$ | 3.31 | 1.05 | 0.87 | 0.0 |
| | $\blacksquare|D_{F_\theta}^{\neg r \wedge a}|$ | 17.19 | 16.87 | 15.96 | 93.03 |
| | $\blacksquare|D_{F_\theta}^{r \wedge a}|$ | 66.08 | 73.85 | 76.53 | 0.0 |
| $\mathcal{B}_{8/255}^\infty$ | $\blacksquare|D_{F_\theta}^{\neg r \wedge \neg a}|$ | 18.17 | 9.55 | 9.21 | 6.97 |
| | $\blacksquare|D_{F_\theta}^{r \wedge \neg a}|$ | 2.64 | 0.76 | 1.31 | 0.0 |
| | $\blacksquare|D_{F_\theta}^{\neg r \wedge a}|$ | 29.79 | 27.82 | 23.78 | 93.03 |
| | $\blacksquare|D_{F_\theta}^{r \wedge a}|$ | 49.40 | 61.87 | 65.70 | 0.0 |

Table 8: `CIFAR-10` robustness-accuracy dataset partitioning. We consider a `TRADES` (Zhang et al., 2019a) trained ResNet-50, adversarially trained WideResNet-28-10 models (Carmon et al., 2019; Gowal et al., 2020), and a standard trained ResNet-50. Adversarially trained models are trained for the respective perturbation region. Each model is evaluated for the indicated $\ell_\infty$ threat model, using 40-step `APGD` (Croce & Hein, 2020).

Further, we also illustrate the robustness-accuracy dataset partitioning on `CIFAR-100` (Krizhevsky et al., 2009). We consider a standard trained WideResNet-28-10 and the adversarially trained WideResNet-28-10 by Rebuffi et al. (2021). Again, the model by Rebuffi et al. (2021) was pretrained for $\varepsilon_\infty = 8/255$ perturbations and then `TRADES` fine-tuned for the respective threat model indicated in Table 9. We again evaluate the robustness-accuracy dataset partitioning for varying $\ell_\infty$ perturbations using 40-step `APGD` (Croce & Hein, 2020), and list the exact size of each data split in Table 9.

Notably, we observe that on the model by Rebuffi et al. (2021), $15.24\%$ of all test samples are robust but inaccurate for $\varepsilon_\infty = 1/255$ perturbations, which is a significantly larger fraction compared to similar models on `CIFAR-10`.

| Threat Model | Data Split | Relative Split Size [%] | |
|---|---|---|---|
| | | Rebuffi et al. (WRN-28-10) | $\mathcal{L}_{std}$ (WRN-28-10) |
| $\mathcal{B}^\infty_{1/255}$ | $\blacksquare\,|D_{F_\theta}^{\neg r \wedge \neg a}|$ | 15.20 | 19.80 |
| | $\blacksquare\,|D_{F_\theta}^{r \wedge \neg a}|$ | 15.24 | 0.03 |
| | $\blacksquare\,|D_{F_\theta}^{\neg r \wedge a}|$ | 7.75 | 60.27 |
| | $\blacksquare\,|D_{F_\theta}^{r \wedge a}|$ | 61.81 | 19.9 |
| $\mathcal{B}^\infty_{2/255}$ | $\blacksquare\,|D_{F_\theta}^{\neg r \wedge \neg a}|$ | 32.75 | 19.82 |
| | $\blacksquare\,|D_{F_\theta}^{r \wedge \neg a}|$ | 8.71 | 0.01 |
| | $\blacksquare\,|D_{F_\theta}^{\neg r \wedge a}|$ | 5.11 | 77.11 |
| | $\blacksquare\,|D_{F_\theta}^{r \wedge a}|$ | 53.43 | 3.06 |
| $\mathcal{B}^\infty_{4/255}$ | $\blacksquare\,|D_{F_\theta}^{\neg r \wedge \neg a}|$ | 30.57 | 19.83 |
| | $\blacksquare\,|D_{F_\theta}^{r \wedge \neg a}|$ | 4.34 | 0.0 |
| | $\blacksquare\,|D_{F_\theta}^{\neg r \wedge a}|$ | 23.16 | 80.02 |
| | $\blacksquare\,|D_{F_\theta}^{r \wedge a}|$ | 41.93 | 0.15 |
| $\mathcal{B}^\infty_{8/255}$ | $\blacksquare\,|D_{F_\theta}^{\neg r \wedge \neg a}|$ | 33.70 | 19.83 |
| | $\blacksquare\,|D_{F_\theta}^{r \wedge \neg a}|$ | 3.91 | 0.0 |
| | $\blacksquare\,|D_{F_\theta}^{\neg r \wedge a}|$ | 26.66 | 80.17 |
| | $\blacksquare\,|D_{F_\theta}^{r \wedge a}|$ | 35.73 | 0.0 |

Table 9: `CIFAR-100` robustness-accuracy dataset partitioning. We consider a standard trained WideResNet-28-10 and the adversarially trained WideResNet-28-10 by Rebuffi et al. (2021), trained for the respective perturbation region considered in each evaluation. Each model is evaluated for the indicated $\ell_\infty$ threat model, using 40-step `APGD` (Croce & Hein, 2020).

