# OpenReview forum: "Avoiding Robust Misclassifications for Improved Robustness without Accuracy Loss"
_ICLR.cc/2022/Conference — ICLR 2022 Submitted_

### Official Review · Reviewer_7Sk1 · 2021-10-30

**Correctness:** 4
**Technical Novelty And Significance:** 3
**Empirical Novelty And Significance:** 3
**Recommendation:** 8
**Confidence:** 3

**Main Review:**

Pros:
- The accuracy-robustness trade-off is a highly relevant issue. The emphasis of this work to improve robustness without sacrificing the accuracy is compelling.
- Very nice and accessible presentation: the goal of the work is well motivated, the technical steps are explained in sufficient depth.
- The implementation of the robust inaccuracy penalty spans both adversarial and certified training.
- The experiments are of appropriate scope, contain sufficient detail for reproducibility, and substantiate the claims about the proposed approach well.
- Although most of the presented technical aspects featured in previous work, their composition seems carefully designed and sufficiently novel.

Cons:
- Although the problem formulation seems novel (Eq. 4), its implementation under both adversarial and certified scenarios of training are closely based on prior work. Similarly for compositional models (Eq. 16). This is appropriately cited, hence does not present a serious issue, but does somewhat limit the novelty.

A more general concern is that the experimental results often do not yet provide as much insight as they potentially could. The showed improvements on one metric often come at a cost of degrading the other. The analysis focuses almost exclusively on the improvement, and I wish the existing trade-off deserved more discussion and analysis in the main text. For example:
- From Fig. 2, the improvement of robust inaccuracy comes at the expense (in part significant) of robust accuracy. Since robust inaccuracy affects only a fraction of samples (cf. Tab. 1), such cost may outweigh the benefits of the proposed regularisation.
- In Tab. 2, $\mathcal{L}_{\text{CRA}}$ may also decrease the robust sample selection w.r.t. the baseline for Cohen et al. Why may this be the case and why is it not the case of Sehwag et al.?

Minor:
* Although this work relates to adversarial robustness, one could relate the issue of robust inaccuracy to model calibration: the model trained with a robust objective may end up overconfident on misclassified examples. The proposed regularisation, in a way, addresses the issue of miscalibration from the perspective of robustness. Nevertheless, I wonder if and how the proposed approach affects the quality of model calibration.
* While I follow the illustrative goal of Fig. 1,  I wonder why is it reasonable to compare the standard model with 2 (!)  decision boundaries to other models with only one.
* Fig. 2 could be made more illustrative (e.g. by reducing the scale of the y-axis);
* Sec. 7.1, certified robustness: It’s at least necessary to summarise the results in the main text, if there is no space to show the results in full.
* The alternative formulation ($\mathcal{L}_{\text{DGA}}$ loss) does not seem to be beneficial. I’d suggest that it is discussed in the appendix, while the allocated space is to address more important concerns.
* Conclusion is too short. It’d be worthwhile to point out existing limitations, and to offer a perspective on how future works may leverage the obtained results.


**Summary Of The Paper:**

This well presented work is motivated by the relevant issue that improving model robustness while maintaining high accuracy can result in robust inaccuracy, i.e. a non-negligible amount of samples get classified incorrectly, but the category remains consistent for their perturbations. Supporting this observation empirically, this work proposes an approach to address this issue.

There are two main technical contributions. The first complements the standard robustness objectives from the literature with a regularisation term penalising robust inaccuracy. The paper provides specific implementations of this term under two setups: empirical (adversarial) and certified robustness. The second contribution is an abstain model: a learned indicator function that decides which samples are to be processed by the robust (but less accurate) classifier, and which ones should default to the non-robust (but accurate) baseline. As a result, this two-stage approach achieves high robustness without compromising the classification accuracy.
The experiments on standard benchmarks further confirm, that the robust model consistently decreases the fraction of robust inaccurate samples (albeit at the expense to robust accuracy).

**Summary Of The Review:**

This work raises and addresses an interesting artefact of robust training, which was disregarded in previous work. The proposed approach seems technically sound, with a few novel elements. Although the analysis could be stonger, it nevertheless provides sufficient evidence for improved accuracy-robustness trade-off over state-of-the-art methods.

---

> ### Author Response · Authors · 2021-11-14
> **Response to Reviewer 7Sk1**
>
> We thank the reviewer for their thorough review and comments. We will work on incorporating the suggestions in our revised version. Please find the answers to the questions below:
>
> > Figure 2 shows that improved (i.e. lower) robust inaccuracy comes at the cost of lower (in part significant) robust accuracy. Such cost may outweigh the benefits of the proposed regularization.
>
> This is why we evaluate this tradeoff by computing the Pareto front of models fine-tuned at different regularization strengths ($\beta$ parameter), rather than reporting a single number used for the downstream task (Table 3). Choosing an appropriate model on this Pareto front is a design choice and depends on the concrete application. An interesting extension would also be to use model cascades, which would start with the left-most model in Figure 2 (i.e., lowest robust inaccuracy but also lowest accuracy), and progressively fall back to models on the trade-off curve.
>
> > In Table 2, why does the robust sample selection decrease w.r.t. the baseline for Cohen et al. but not for Sehwag et al.?
>
> This is an interesting observation -- we do observe some cases where both robust accuracy and robust selection improve (also Gowal et al., in Figures 3, 7 (blue circles)). However, we do not currently have a concrete justification for why this is the case.
>
> > While I follow the illustrative goal of Figure 1, I wonder why it is reasonable to compare the standard model with 2 decision boundaries to other models with only one?
>
> Note that all the decision boundaries shown in Figure 1 are not selected by us, these correspond to the actual decision boundaries obtained by training each model. Concretely, we use a simple four-layer fully connected neural network and the respective loss functions shown in Figure 1. Technically, all the models have two decision boundaries, it just so happens that one decision boundary, obtained by TRADES and our method, is so small that it is not visible in the visualization. We do however see that this can be confusing and we will clarify this in the text.
>
> > It would be beneficial to move the alternative formulation ($\mathcal{L}_{DGA}$ loss) to the appendix and use the space to extend the conclusion.
>
> Thank you for the suggestion. We will revise our work accordingly.

---

> > ### Comment · Reviewer_7Sk1 · 2021-11-19
> > **Response to authors**
> >
> > I thank the authors for their response and some additional clarifications. While I still lean positive on this work, I agree with the other reviewers that both the technical presentation and result analysis should be made clearer and more accessible. I hope, we are yet to see a revised version of the manuscript, which can potentially address most of the raised issues.

---

> > > ### Author Response · Authors · 2021-11-20
> > > **Response**
> > >
> > > Thank you for the response. We are working on a revised version of our paper, it will be ready before the second stage begins (22.11).

---

### Official Review · Reviewer_sDbQ · 2021-11-01

**Correctness:** 4
**Technical Novelty And Significance:** 2
**Empirical Novelty And Significance:** 2
**Recommendation:** 5
**Confidence:** 3

**Main Review:**

1. It's interesting to improve robust training methods by avoiding robust inaccuracy, but the proposed method is a little bit straightforward.
2. I think more discussion and analysis should be provided to explain the reason behind the effectiveness of the proposed method, empirically or theoretically. To be clear, I'm not requesting the authors to provide theoretical justification, and just suggest providing more insights and analysis.
3. The Figure2, 3, 4 can be improved, and I was confused about what did the curves represent at first.

**Summary Of The Paper:**

To tackle the two limitations of the current robust training algorithms: 1). robust inaccuracy 2). the sacrifice of natural accuracy, this paper proposed a new training method that aims to maximize robust accuracy and minimize robust inaccuracy. Moreover, a robustness-based abstain mechanism is adopted to further boost overall robustness without sacrificing accuracy. Experiments show the effectiveness of the algorithm in terms of fewer robust and inaccurate samples and better robustness, with only marginally reduced natural accuracy.

**Summary Of The Review:**

From my perspective, the proposed approach is straightforward and not fully explained. Thus I tend to reject this work. But I am open with my score, according to the authors' responses and other reviewers' comments.

---

> ### Author Response · Authors · 2021-11-13
> **Response to Reviewer sDbQ**
>
> We thank the reviewer for their comments.
>
> > Figures 2, 3, 4 can be improved, and I was confused about what the curves represent at first.
>
> We have tried hard to make the figures intuitive. In particular, we’ve made sure that we can easily differentiate three main components of our experiments by making them visually distinct:
>
> - *shape filling* was used to differentiate models trained with and without augmentations by labeling them using empty and full markers, respectively.
> - *marker type* was used to differentiate model types
> - *color* was used to differentiate our work vs prior work.
>
> We also incorporated the markers inline in the text, to make the connection explicit. Having said that, we are keen to incorporate more feedback and discuss more ideas that could improve our presentation.
>
> Given that plots showing robust inaccuracy vs robustness are a unique aspect of our work and not found in other papers, it does make them somehow harder to understand at first. However, we do believe such plots have high information value, especially that they give the Pareto front of solutions and allow comparing different techniques as a whole (which is not possible with a table in this case).
>
> > Can you provide more discussion to explain the reason behind the effectiveness of the proposed method?
>
> We have included some intuition behind our method in Figure 1. It shows a synthetic example that illustrates deficiencies of both standard training and adversarial training. The effectiveness of our work lies in combining the strengths of both by ensembling natural and robust models into one, with the goal of retaining the accuracy while improving the robustness. The key challenge lies in how these models should be combined, which we address by using robustness as a principled selection (abstain) mechanism. We hypothesize that empirically, the robust inaccurate samples lie in regions that are close by in the learned manifold, thus regularizing them to become even slightly non-robust generalizes well.

---

### Official Review · Reviewer_wtee · 2021-11-03

**Correctness:** 3
**Technical Novelty And Significance:** 3
**Empirical Novelty And Significance:** 3
**Recommendation:** 5
**Confidence:** 3

**Main Review:**

The abstract assumes the reader is aware that the paper will deal with adversarial DL, it would be better to "set the stage" a bit.
Without giving the setting of robustness to adversarial examples, the abstract is not clear to me, as it is using a lot of terms and concepts that will only become clear during the paper, such as: "robust inaccuracy", "$\epsilon_\infty$ robustness" and "certified robustness".

Table 1 seems to be a justification why the approach presented in the paper is needed. However not much details are given on how to interpret the results shown and how the values were generated.

The datasets are only explained & cited in section 7, however the abbreviations are used already in the introduction. Either cite them correctly and declare the abbreviations at first occurrence or do not mention them explicitly in the introduction.

Instead of mixing experiments (on synthetic data) and partial method descriptions with the introduction, i would favor a more detailed explanation of the introduced concepts and presented experiments. In the introduction i would suggest to focus on the gap in previous works, while keeping the experimental investigations for the respective chapter and providing more details there.

In the related work section there are several claims and properties of the proposed approach mentioned, that are not yet explained or proven, i would be better to at least point the reader to where that will be done.

The important concepts of "natural accuracy", "robust accuracy" and "robust inaccuracy" are only explained in section 3 but used to a large extend prior to that section. At least reference the explanation coming at a later stage or consider rearranging the order by which concepts are introduced. First introduce the concept than use it to explain the novelty or justification of the proposed approach.

Minor point, but similar issue: indicator function introduced in section 4, but already used in section 3.

The compositional architecture seems like ensembling with weighting based on selector S. It would be helpful to relate to that.

Illustration in figure 2 seems rather cluttered, difficult to grasp the gain by the proposed method.

## Questions to the authors
Have you compared fine-tuning existing models with training them from scratch with the proposed loss?
Why would the method be restricted to fine-tuning?

**Summary Of The Paper:**

The paper addresses the issue of trained models being inaccurate but robust for some samples. To address this issue a flexible fine-tuning mechanism is proposed. A robustness-based ensembling method is introduced as well.

**Summary Of The Review:**

In summary a few things are a bit unclear, e.g. to me why the method would be restricted to fine-tuning. Can certainly be improved by re-structuring, some suggestions given above. In the current state for me a bit hard to grasp the contribution.

---

> ### Author Response · Authors · 2021-11-13
> **Response to Reviewer wtee**
>
> We thank the reviewer for their questions and comments. We will work on incorporating the suggestions in our revised version. Please find the answers to the questions below:
>
> > Is the method restricted to fine-tuning?
>
> No, our method is not restricted to fine-tuning and can be used to train the model from scratch. To obtain the best results, it is beneficial to introduce a scheduler for the $\beta$ parameter, that controls how much the model is regularized (penalized) for robust inaccurate samples. When the model is trained from scratch, it is best to start with small (or even no) regularization and then increase it as the training progresses.
>
> > Have you compared fine-tuning existing models vs training them from scratch with the proposed loss?
>
> As outlined above, our current experiments can be thought of as using a binary scheduler that starts without any regularization and then transitions to regularization with strength $\beta$ after $n$ epochs. While it is likely that different schedulers would improve the results, we have so far only focused on the case of fine-tuning existing models.
>
> > Can you give more details on Table 1, how to interpret the results and how the results were evaluated?
>
> Yes, Table 1 can be summarized as:
>
> *What is shown*: the ratio of robust inaccurate samples across multiple datasets, models (columns), and perturbation regions (rows).
>
> *How were the models trained*: for CIFAR-10 and CIFAR-100, we evaluate existing pre-trained models from Zhang et al., Carmon et. al., Gowal et al. and Rebuffi et al. For MSTD (mapillary traffic sign dataset) and SBB (rail defect dataset), we trained models from scratch using TRADES [1], as pre-trained robust models for these datasets were not available.
>
> *How were the models evaluated*:  using 40-step APGD [2]. Note that our evaluation in fact computes the breakdown of all 4 metrics - robust accuracy, robust inaccuracy, non-robust accuracy, non-robust inaccuracy (e.g., see Table 6 in the appendix).
>
> *Results interpretation*: samples that are robust but inaccurate can be dangerous if misinterpreted -- the robustness gives a false sense of security that the model is performing well and is safe to use. What the results show is the severity of this issue for existing models and datasets. For example, for the $\mathcal{B}_{1/255}^{\infty}$ threat model, 15% of CIFAR-100 samples are robustly misclassified as an incorrect class. Similarly, 7% in the rail defect dataset are robust but incorrect.
>
> [1] Zhang, H., Yu, Y., Jiao, J., Xing, E., El Ghaoui, L., & Jordan, M. (2019, May). Theoretically principled trade-off between robustness and accuracy. In International Conference on Machine Learning (pp. 7472-7482). PMLR.
>
> [2] Croce, Francesco, and Matthias Hein. "Reliable evaluation of adversarial robustness with an ensemble of diverse parameter-free attacks." International conference on machine learning. PMLR, 2020.

---

> > ### Comment · Reviewer_wtee · 2021-11-30
> > **Response to the authors**
> >
> > I'd like to thank the authors for the response and the updated version of the paper. I would like to acknowledge that the structure/presentation of the paper structure have been improved.

---

### Official Review · Reviewer_xSkN · 2021-11-03

**Correctness:** 3
**Technical Novelty And Significance:** 1
**Empirical Novelty And Significance:** 2
**Recommendation:** 3
**Confidence:** 4

**Main Review:**

I found it difficult to understand this paper. The goal is not well motivated, the proposed terminology is confusing, and I'm not sure what the takeaway is from the results.

First of all, the abstract mentions numbers without sufficient context. It is not even stated which benchmark this sentence refers to: "natural accuracy from 97.8% to 97.6%". After going through the paper, one can infer that they were referring to CIFAR-10, but even then the model name and training settings are not mentioned (either on table 3 or section 7.3). I see that one could follow the reference or the appendix, but basic information such as this one should be included in the main text.

Similarly, the phrase "robust inaccuracy" is not defined until page 3, even though it is mentioned in the abstract and the introduction. I would suggest authors to define robust inaccuracy, even if informally, earlier in the paper. My personal guess is that a phrase like "mistake consistency" or "prediction consistency" would be less confusing than the phrase "robust inaccuracy".

I would also suggest the authors try to connect the "robust inaccuracy" aspect of their work with previous work. The related work section mainly touches on the "abstain option" and the "robustness vs accuracy tradeoff". Regarding the "robust inaccuracy", I'm not sure if this concept as been studied with that name before. But relevant work includes prediction consistency of models in semi-supervised learning and adversarial robustness of models trained on random labels. It is surprising (and relevant to this paper) that models trained on random labels have similar behavior w.r.t. adversarial attacks as models trained on regular data. Thus the adversarial robustness on a sample may not depend too much on whether the sample itself is classified correctly or incorrectly by the model.

My biggest concern about this submission is that I do not see the motivation of studying the "robustness of inaccuracy" on samples that are already classified incorrectly. The adversarial direction is defined as the direction that increases the loss, so it's not surprising that the predictions are still incorrect after the adversarial direction is added to the sample. An interesting study could look at the dynamics of going from one incorrect prediction to another one, but that is not explored in this submission.



**Summary Of The Paper:**

This submission aims to reduce the prevalence of samples for which a neural network might predict the wrong answer, not just for the sample but also its nearby points in the input space. Authors call this metric robust inaccuracy.

Next, after reducing the robust inaccuracy metric, they take advantage of their models to improve conventional robustness, by taking advantage of their more reliable robustness metric to abstain from samples for which model prediction is not robust (perhaps better to call this consistent, since it can refer to both correct and incorrect predictions).

**Summary Of The Review:**

I found that the goal of the paper is not well motivated and the presentation should be improved before it can be accepted.

---

> ### Author Response · Authors · 2021-11-13
> **Response to Reviewer xSkN**
>
> We thank the reviewer for their interesting questions and comments. We first discuss the key concerns, which we believe can be addressed by clarifying the main goal of our work. We then also include answers to minor clarifications pointed out by the review:
>
> *Key Concerns:*
>
> > I do not see the motivation of studying the “robustness of inaccuracy” on samples that are already classified incorrectly.
>
> Quite the opposite, imagine a medical model that predicts cancer from an MRI scan. At inference time, while it is not known whether the model prediction is correct (as the ground truth is not available), it is possible to assess the model robustness. This is where the samples that are robust but inaccurate are extremely dangerous -- the robustness gives a false sense of security that the model is performing well and is safe to use.
>
> In other words, our work is important for all the domains that would aim to use the model robustness as a measure of model quality. This is in addition to the robustness and accuracy benefits we illustrate in our evaluation.
>
> > If the sample is already incorrect, it is not surprising that the predictions are still incorrect after the adversarial direction is added to the sample.
>
> While this would indeed not be surprising, this is not what we do in our work.
>
> Instead, robustness is always evaluated without the knowledge of the ground truth label and only using the information available at inference time (see Equations 2 and 3 in the paper). Formally:
> $$\mathbf{1} \\{ \forall \boldsymbol{x}' \in \mathcal{B}_{\epsilon}^{p}(\boldsymbol{x}).~F_\theta(\boldsymbol{x}') = F_\theta(\boldsymbol{x}) \\}$$
> As can be seen, this formulation never uses the ground truth label ($y$), only the model prediction at the input $F_\theta(\boldsymbol{x})$ and all the valid perturbations $F_\theta(\boldsymbol{x}')$.
>
>
> *Minor Clarifications:*
>
> > Can you update the abstract to explicitly mention the experiment context (models and datasets)?
>
> Thank you for the suggestion, we are happy to make this more explicit.
>
> > Would a phrase like "mistake consistency" or "prediction consistency" be more fitting than the phrase "robust inaccuracy"?
>
> We thought about the name “robust inaccuracy”, as well as “consistency”, as it is indeed quite intuitive. At the time of writing, we decided on "robust inaccuracy" as it combines two terms that already exist and reflect technically what is being checked. Essentially:
>
> - Standard training considers: accuracy, inaccuracy (mispredictions)
> - Adversarial training considers: robustness, non-robustness
> - Our work considers all four combinations, with the main focus on robust inaccuracy.
>
> |        |    Accuracy       | Inaccuracy  |
> | -------------: |:-------------| :-----|
> | Robustness    | Adversarial Training | Our Work |
> | Non-robustness| Standard Training |   Our Work |
>
> We thought using two existing terms would be less confusing and more intuitive.
>
> > Can you relate robust inaccuracy to recent works examining adversarial robustness of models trained on random labels? These works suggest that adversarial robustness on a sample may not depend too much on whether the sample itself is classified correctly or incorrectly by the model.
>
> This is an interesting recent work (e.g., concurrent paper to our work [1], also under review at ICLR’22), which in fact supports the usefulness and the motivation of our work.
> Indeed, the observation that both correct and incorrect samples can be robust is at the core of our technique -- these are robust because the model was trained to be robust on them and has enough capacity to achieve this. Instead, using our method, the robust inaccuracy is minimized by explicitly encoding it as part of the optimization objective.
>
> > I would suggest defining robust inaccuracy, even if informally, earlier in the paper.
>
> We are happy to incorporate this suggestion in the revised version of our work.
>
>
> [1] Dong, Yinpeng, Ke Xu, Xiao Yang, Tianyu Pang, Zhijie Deng, Hang Su, and Jun Zhu. "Exploring Memorization in Adversarial Training." arXiv preprint arXiv:2106.01606 (2021).

---

### Author Response · Authors · 2021-11-23
**Paper Revision**

Dear reviewers, we would like to thank you for all your comments and suggestions.
We have uploaded a revised version of our work that incorporates various suggestions and minor clarifications based on your reviews.

For convenience, we summarize the main changes below:

[Abstract] We added context to the results presented in the abstract, i.e. which dataset is considered and how is the model trained on which we improve on.

[1 Introduction] We have modified the introduction to better introduce certain concepts, give more context where needed, and refer readers to sections in the paper where more context is given. Specifically:
- We added an informal definition of “robust inaccuracy”, and we refer the reader to the formal definition of “robust inaccuracy” in Section 3.
- We added more context into how the results in Table 1 were obtained and reference the reader to Section 7, where the considered models and datasets are described in more detail.
- We added clarifications for Figure 1 and references to the hyperparameters (Appendix A.2) that were used to generate and train on the synthetic dataset.

[3 Preliminaries] We changed the robust accuracy abbreviation to  $\mathcal{R}_{rob}^{acc}$, in order to clearly differentiate from the robust inaccuracy abbreviation.

[5 Robust Abstain Models] The alternative loss formulation $\mathcal{L}_{DGA}$ was moved to Appendix A.3.

[7 Evaluation] We incorporated various comments and suggestions to improve the presentation of our results. Concretely:
- We indicate the direction of improvement in plots, making interpretation of the results easier.
- Figure 2: We decluttered Figure 2 by only showing results for one single perturbation region ($\epsilon_\infty = 2/255$), increasing the size of plots, and moving additional results into Appendix A.6.
- Figure 3: We switched the x- and y-axis (robust selection on x-, robust accuracy on y-axis), which is now more consistent with the rest of the paper.
- Section 7.1: We have moved the results on reducing robust inaccuracy for certifiably robust models from the appendix to Section 7.1.

[8 Conclusion] We rewrote the conclusion to be more extensive. We summarize the presented work, highlight limitations, and propose possible future work items.

[Appendix A.2] We added hyperparameters for training from scratch using TRADES, and hyperparameters used to generate and train on the synthetic dataset (Figure 1).

[Appendix A.3] We moved the additional loss formulation $\mathcal{L}_{DGA}$ from Section 5 to A.3.

[Appendix A.5] We added Figure 7, which shows additional robust accuracy/robust inaccuracy plots (for $\epsilon_\infty = 1/255$), which were removed from Figure 2.

[Appendix A.6] We switched the x- and y-axis in Figure 8 (robust selection on x-, robust accuracy on y-axis), which is now more consistent with the rest of the paper.

---

### Decision · Program_Chairs · 2022-01-20

**Decision:**

Reject

**Comment:**

This paper presents the problem of robust inaccuracy (model predictions being robust to perturbations but inaccurate on datapoints), and present methods to maximize robustness while avoiding robust inaccuracy. Furthermore, they develop an abstention mechanism based on robustness to prevent prediction on points where the model is not robust. Results show improvement in adversarial robustness to standard attacks with only small reduction in natural accuracy.

Reviewers were mixed on the clarity and importance of this submission. A major concern raised was on the importance of robust inaccuracy, motivation for avoiding it, and novelty of the proposed method. Other abstention mechanisms are available and one does not solely need to rely on robustness. Additionally, results are often presented on a pareto front and the method does not strictly dominate prior approaches. Authors addressed many of the clarity concerns in their updated revision, and reviewers commented on the high quality of analysis performed in the experiments. But several reviewers still found the draft and description of the robust inaccuracy problem insufficiently motivated and the methodology not well explained. Given lingering concerns over clarity and motivation (in spite of a revision that exceeds the page limit), I cannot recommend this paper for acceptance.